# Degassing-induced fractionation of multiple sulphur isotopes unveils post-Archaean recycled oceanic crust signal in hotspot lava

Patrick Beaudry[1,5], Marc-Antoine Longpré[1,2], Rita Economos[3], Boswell A. Wing[4,6], Thi Hao Bui[4] & John Stix[4]

Mantle source regions feeding hotspot volcanoes likely contain recycled subducted material. Anomalous sulphur (S) isotope signatures in hotspot lavas have tied ancient surface S to this deep geological cycle, but their potential modification by shallow magmatic processes has generally been overlooked. Here we present S isotope measurements in magmatic sulphides, silicate melt inclusions and matrix glasses from the recent eruption of a hotspot volcano at El Hierro, Canary Islands, which show that degassing induces strongly negative $\delta^{34}S$ fractionation in both silicate and sulphide melts. Our results reflect the complex interplay among redox conditions, S speciation and degassing. The isotopic fractionation is mass dependent ($\Delta^{33}S = 0‰$), thus lacking evidence for the recycled Archaean crust signal recently identified at other hotspot volcanoes. However, the source has an enriched signature ($\delta^{34}S$ ~ + 3‰), which supports the presence of younger $^{34}S$-rich recycled oceanic material in the Canary Island mantle plume.

[1] School of Earth and Environmental Sciences, Queens College, City University of New York, Queens, NY 11367, USA. [2] Earth and Environmental Sciences, The Graduate Center, City University of New York, New York, NY 10016, USA. [3] Department of Earth Sciences, Southern Methodist University, Dallas, TX 75275, USA. [4] Department of Earth and Planetary Sciences, McGill University, Montréal, QC H3A 0E8, Canada. [5]Present address: Department of Earth, Atmospheric and Planetary Sciences, Massachusetts Institute of Technology, Cambridge, MA 02139, USA. [6]Present address: Department of Geological Sciences, University of Colorado Boulder, Boulder, CO 80309, USA. Correspondence and requests for materials should be addressed to P.B. (email: pbeaudry@mit.edu)

Chemical, physical and biological processes within the mantle, crust, atmosphere and oceans fractionate sulphur isotopes, and material exchanges among these geological reservoirs lead to characteristic sulphur isotope signatures that have varied over time[1]. Sulphur isotope heterogeneity in the mantle, as sampled by sulphide inclusions[2–5], melt inclusions[6,7] or primitive volcanic rocks[8,9], thus traces secular variation in the tectonic boundary conditions that influence mantle circulation, as well as the internal processes that have established the current mantle state. On the other hand, the full scope of mantle heterogeneity is best illustrated by variations in trace element and radiogenic isotope geochemistry of ocean island basalts (OIB), which define various mantle end-members thought to feed the sources of hotspots[10,11]. Therefore, it is of interest to investigate these mantle components from a sulphur isotope perspective to help set additional constraints on their origin and further our understanding of the deep sulphur cycle. This type of approach has been adopted in recent studies, which we summarise below.

Mass-independent fractionation (MIF) of sulphur isotopes is defined by non-zero $\Delta^{33}S$ values, where $\Delta^{33}S = \delta^{33}S – [(1 + \delta^{34}S)^{0.515}–1]$ and $\delta^{x}S_{V\text{-}CDT} = (^{x}S/^{32}S)_{sample}/(^{x}S/^{32}S)_{V\text{-}CDT} – 1$, and is characteristic of sedimentary rocks of Archaean age (~2.5 Ga and older), reflecting the influence of photochemical processes in an atmosphere devoid of oxygen[12]. Sulphur in sedimentary rocks that post-date the Great Oxidation Event at ~2.4 Ga thus has $\Delta^{33}S$ values around 0‰, and the associated S isotope fractionations are termed mass-dependent. However, negative, non-zero $\Delta^{33}S$ values have now been reported twice for young volcanic rocks from hotspot settings: in olivine-hosted sulphides from Mangaia, Cook Islands[4], and in sulphides from the Pitcairn hotspot[5]. These anomalous signatures are thought to reflect the cycling of Archaean sulphur from Earth's surface to the mantle by subduction, and back to the surface via mantle plumes[4,5,13]. Since Mangaia is the representative end-member of the ancient 'HIMU' (high $\mu = {}^{238}U/{}^{204}Pb$) mantle component[10,11], the potential positive covariation of $\Delta^{33}S$ and $\delta^{34}S$ values in Mangaia sulphides was first used to suggest a specific Archaean protolith to the HIMU source characterised by negative $\Delta^{33}S$ and $\delta^{34}S$ values[4]. The subsequent finding of S-MIF at Pitcairn, representative of the enriched mantle I end-member[10,11] (EM-I; characterised by unradiogenic Pb isotope signatures), in association with negative $\delta^{34}S$ values, lends support to this hypothesis, potentially resolving the positively-skewed imbalance of $\Delta^{33}S$ values observed in Archaean surface reservoirs[5,13]. These studies thus imply that a missing Archaean sulphur pool is stored in the deep mantle and occasionally resurfaces at hotspots. Other plume-related lavas from Samoa, the type locality for the third common OIB mantle isotopic end-member, EM-II (characterised by the highest ${}^{87}Sr/{}^{86}Sr$ ratios), show coupled variations in S and Sr isotopes that indicate recycling of younger sulphur-rich sediments into a mantle source with a near-zero $\Delta^{33}S$ value (mass-dependent) and a $\delta^{34}S$ value of ~3‰[9]. The distinction in S isotope signatures between different mantle reservoirs suggests a long-lived and isotopically evolving surficial input into different hotspot source regions, highlighting the importance of understanding the causes of S isotope variability in the mantle.

Magmatic processes involving sulphur, such as degassing, sulphide segregation, mixing or assimilation, may also leave an imprint on the S isotope composition of volcanic rocks[7,8,14,15], and separating these effects from the source signature can present challenges. However, melt inclusions, which represent droplets of silicate melt trapped in minerals during crystal growth, offer unique snapshots of an ascending magma at various depths[16], and can help resolve this problem. Toward this end, we report the S isotope compositions of melt inclusions and matrix glasses

spanning the degassing history of the volatile- and sulphur-rich magma erupted at El Hierro, Canary Islands, in 2011–2012[17]. We also present $\delta^{34}S$ and $\Delta^{33}S$ signatures of magmatic sulphide inclusions — to our knowledge, this is the first time S isotopes are measured simultaneously in sulphides, melt inclusions and matrix glass for a single eruption. The dataset provides an unusually clear picture of the mechanisms by which sulphur isotopes fractionate during degassing and sulphide saturation in natural magmas, and shows that degassing can induce large $\delta^{34}S$ fractionations of up to 10‰. In turn, this offers an exceptional opportunity to investigate in situ how the S isotope heterogeneity generated during magma evolution, ascent and eruption[7,8,18,19] can be quantitatively discriminated from that inherited from the mantle sources for hotspot volcanism. In doing so, we find that the S isotope signals at El Hierro reflect a post-Archaean origin for recycled S in the Canary Island hotspot, contrasting with the recent findings of S-MIF at other OIB localities.

## Results

**Geological setting and sample description.** The Canary Island hotspot in the eastern Atlantic Ocean is characterised by an exotic geochemistry, producing OIB that is mostly alkaline in composition and displays isotopic affinities with the HIMU, EM and depleted MORB mantle (DMM) end-members[20,21], suggesting that the mantle source constitutes a mix of different reservoirs[21]. This context thus offers the potential to confirm or contrast S isotope signals from the Canary Islands to those observed at hotspots in the South Pacific (i.e. Mangaia/HIMU; Pitcairn/EM-I; Samoa/EM-II). Additionally, ubiquitous fluid inclusions in mantle xenoliths and occurrence of carbonatite melt[22,23] point to a mantle source enriched in volatiles. The recent submarine eruption off the south coast of El Hierro, the youngest and westernmost island of the archipelago, produced lava balloons containing olivine-hosted melt inclusions (Fig. 1a) with dissolved volatiles reaching concentrations in excess of 3000 ppm $CO_2$, 3 wt.% $H_2O$ and 5000 ppm S (ref. [17]). In addition, clinopyroxene and spinel (Fe–Ti oxide) phenocrysts in the same samples host abundant sulphide globules (Fig. 1b–f), revealing that the magma was saturated with an immiscible sulphide liquid for part of its history. However, sulphide inclusions are not present in olivine phenocrysts, nor do they occur as a free phase in the matrix glass.

We performed in situ sulphur isotope analyses by Secondary Ion Mass Spectrometry (SIMS) on a suite of 25 olivine- and spinel-hosted melt inclusions and 9 matrix glass chips, previously shown to have large ranges in volatile contents and sulphur speciation, with S contents linearly and positively correlated with $H_2O$ and $S^{6+}/\Sigma S$ (ref. [17]). These measurements yielded $\delta^{34}S$ values only, owing to the analytical difficulty of resolving low-abundance ${}^{33}S$ at low levels of S in silicate melts. We also obtained the S isotope compositions ($\delta^{34}S$ and $\Delta^{33}S$) of 49 clinopyroxene- and spinel-hosted sulphide droplets, and measured their chemical composition by electron probe micro-analysis (EPMA). Table 1 summarises the isotopic composition of the various samples. Details on our analytical techniques and associated uncertainties can be found in the Methods section. The Supplementary Information includes a discussion of potential matrix effects during isotopic analysis, which appear negligible. All quoted errors are $1\sigma$ propagated analytical uncertainties (see Methods for error treatment).

**Sulphur isotopes in the silicate melt.** The $\delta^{34}S$ values of melt inclusions and matrix glasses (Supplementary Data 1) exhibit an exponential decline with decreasing S content (Fig. 2a), with S-rich inclusions having positive $\delta^{34}S$ values and S-poor matrix glasses (<500 ppm S) having the most negative $\delta^{34}S$ values.

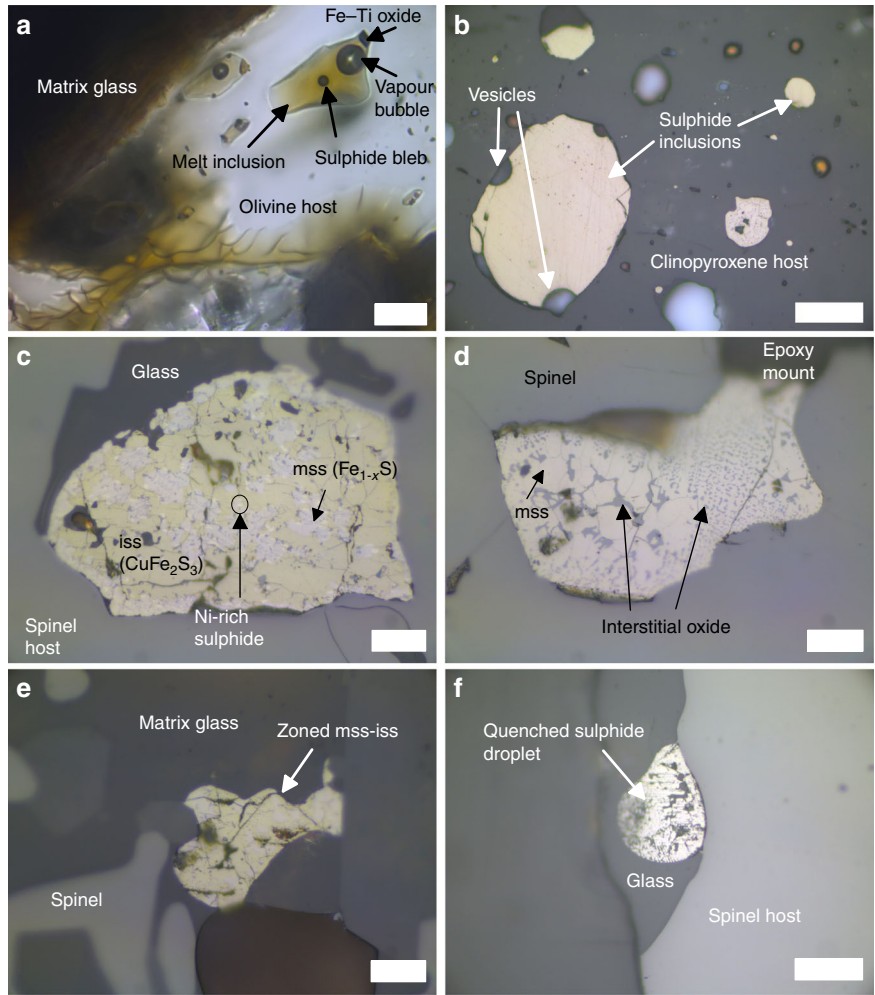

**Fig. 1** Inclusions analysed in situ by Secondary Ion Mass Spectrometry (SIMS). **a** Plane-polarised light photomicrograph of an olivine-hosted melt inclusions with exsolved bubbles. **b** Reflected light photomicrograph of homogeneous, rounded monosulphide solid solution (mss) sulphides of pyrrhotite composition hosted in a clinopyroxene phenocryst (group 1a). **c** Sub-angular sulphide inclusion with intergrowths of mss and intermediate solid solution (iss) (group 2), as well as small exsolutions of a Ni-rich phase at their interface (see elemental maps in Supplementary Fig. 3). Note the contact between the inclusion and silicate melt. **d** Irregular shaped inclusion of mss with interstitial oxide (group 2). **e** Group 2 zoned mss-iss sulphide surrounded by melt and a skeletal, rapidly growing spinel phenocryst. **f** Rounded sulphide droplet, with trellis texture and attached to the outside of an spinel phenocryst, also in contact with melt. Scale bars are 20 μm for panels **a**, **c**–**e** and 50 μm for panels **b**, **f**

| Table 1 Summary of samples analysed with ranges in S content and isotope composition | | | | | |
|---|---|---|---|---|---|
| Type of sample | Number of samples | δ³⁴S max (‰) | δ³⁴S min (‰) | S content max | S content min |
| *Silicate melt* | | | | | |
| Olivine-hosted melt inclusions | 15 | 3.2 | −5.9 | 5080 ppm | 520 ppm |
| Spinel-hosted melt inclusions | 10 | 3.1 | 0.8 | 4010 ppm | 1760 ppm |
| Matrix glass chips (averages) | 9 | −0.9 | −8.2 | 570 ppm | 370 ppm |
| Matrix glass individual analyses | 17 | 0.4 | −9.6 | NA | NA |
| *Sulphide melt* | | | | | |
| Group 1 sulphides (mss) | 33 | 1.0 | −4.0 | 39.1 wt.% | 37.7 wt.% |
| Group 2 sulphides (mss-iss)ᵃ | 12 | −1.8 | −7.1 | 37.9 wt.% | 33.0 wt.% |
| Group 3 sulphides (quenched) | 4 | −2.3 | −9.6 | 33.4 wt.% | 31.4 wt.% |

ᵃThe ranges for group 2 are those measured in mss, for comparison with groups 1 and 3 sulphides

Spinel-hosted silicate melt inclusions have $\delta^{34}S$ ranging from $+3.1 \pm 0.5‰$ to $+0.8 \pm 0.5‰$ while olivine-hosted melt inclusions span a greater range from $+3.2 \pm 0.1‰$ to $-5.9 \pm 0.6‰$. Matrix glasses have the lowest $\delta^{34}S$ values, extending down to $-8.2 \pm 0.6‰$ (Fig. 2a). Inclusion size (30–150 μm) show no relationship with isotopic composition.

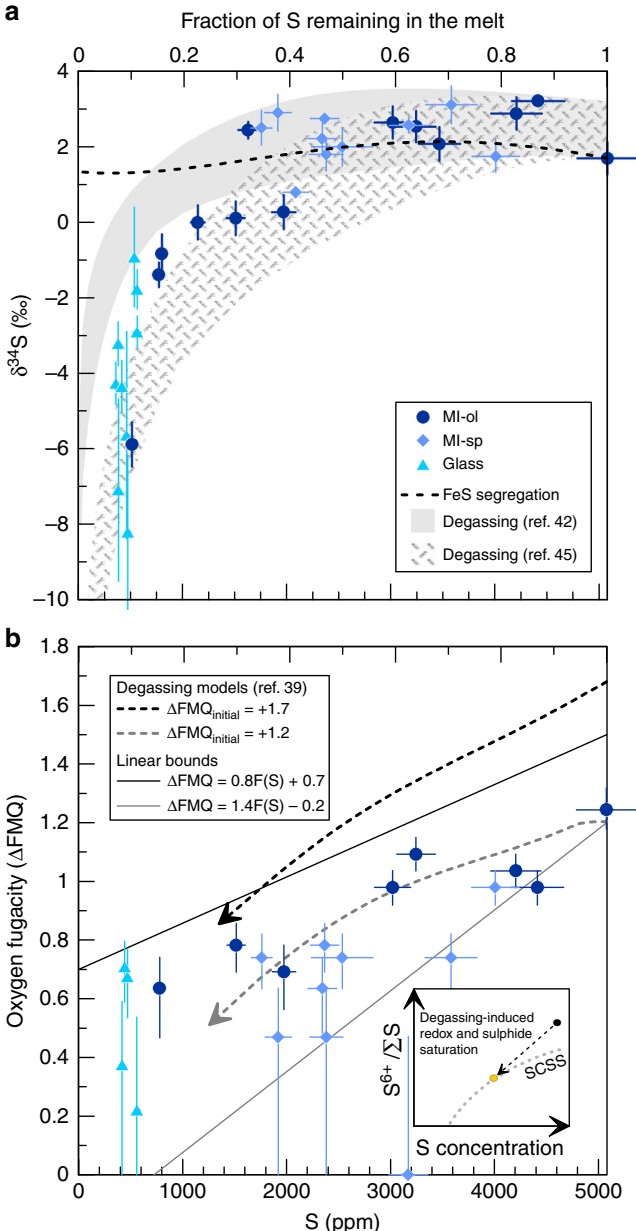

**Fig. 2** Degassing-induced sulphur isotope fractionation. **a** Sulphur content and isotopic composition measured in olivine-hosted (MI-ol; navy blue circles) and spinel-hosted (MI-sp; blue diamonds) melt inclusions and matrix glasses (light blue triangles). Shaded regions represent the possible ranges of isotopic fractionation due to degassing, with changing pressure and oxygen fugacity, computed with the model of Marini et al.[14] with the empirical fractionation factors of Miyoshi et al.[42] (grey fill) and Fiege et al.[45] (hatched field). Minimum and maximum starting $\delta^{34}S$ values are chosen as the $\delta^{34}S$ value of the most S-rich inclusion and the maximum $\delta^{34}S$ value, respectively. The lower and upper bounds for each field correspond to S-$fO_2$ covariation as shown by the grey and black solid lines in b, with varying pressure as modelled in Supplementary Fig. 5. The dashed line shows the effect of FeS segregation on melt $\delta^{34}S$ values. Error bars represent $1\sigma$ propagated analytical uncertainties (see Methods). **b** Melt oxygen fugacity ($\Delta$FMQ: log $fO_2$ units relative to the fayalite–magnetite–quartz buffer) as a function of S content (F(S)), derived from $S^{6+}/\Sigma S$ ratios of melt inclusions and matrix glass with the equation of Jugo et al.[33]. Solid lines are linear relations drawn to encompass the data, representing minimum $fO_2$ (grey line) and maximum $fO_2$ (black line) for a given S content, and are used for modelling the lower and upper-bound fractionation estimates for each model shown in a. Dashed arrows represent degassing paths calculated with D-Compress (ref. [39]) in the C–S–O–H–Fe system, starting at a pressure of 300 MPa, initial volatile contents as measured in the most volatile-rich inclusion, and minimum (grey arrow) and maximum (black arrow) estimated initial $fO_2$. The inset shows the expected trajectory of the magma, with initially high $S^{6+}/\Sigma S$ ratio (black dot) decreasing upon S degassing (black dashed arrow) until sulphide saturation is reached (yellow dot). At that point the magma follows the SCSS upon further degassing (grey dotted line). Error bars for $\Delta$FMQ represent the analytical uncertainty on $S^{6+}/\Sigma S$ values, converted to $\Delta$FMQ values[17,33]

and 0.4–1.9 wt.% Cu; Fig. 4a and Supplementary Data 2), and sometimes contains small exsolutions of a more Ni-rich phase, likely pentlandite (Fig. 1c, Supplementary Fig. 3b). The iss has a stoichiometry corresponding to cubanite (CuFe$_2$S$_3$) (Fig. 4a). The oxide phase was too fine-grained to analyse by EPMA, but elemental maps obtained with a scanning electron microscope (SEM) confirmed that its composition is essentially iron oxide (Supplementary Fig. 4). These inclusions are hosted primarily in spinel phenocrysts, commonly occur at the edge of crystals, and are sometimes in contact with the melt or even completely outside the crystal (Fig. 1e). They are also often associated with vesicles (cf. ref. [27]). They have $\delta^{34}S$ ranging from −1.8‰ to −7.1‰ (Figs 3 and 5).

Four additional globules have spherical shapes resembling the homogeneous inclusions of group 1, but have trellis-textured surfaces, which are often observed for quenched sulphide melt[28,29], and also occur outside or at the edge of crystal hosts — we distinguish them as group 3 (Fig. 1f). They are characterised by the highest Fe and lowest S contents, as well as low Ni and Cu (Supplementary Data 2, Figs 4, 5). Their $\delta^{34}S$ range is similar to that of the group 2 sulphides, from −2.3‰ down to −9.6‰. However, they do not contain a discrete oxide phase as seen in group 2, and have low EPMA element totals (~95 wt.%). This may be due to incorporation of $O^{2-}$ anions in the sulphide liquid[30,31], given that oxygen was not included in the analytical routine. One of these inclusions also has an interstitial Cu-rich phase (Supplementary Fig. 3e).

Sulphides analysed from all three groups have similar size ranges from 25 μm up to 180 μm across, and there is no systematic variation in isotopic composition or geochemistry with size. Smaller droplets (<10 μm) were also present but not analysed due to the ~15 μm size of the ion beam (see Methods).

**Sulphur isotopes and petrography of the sulphides.** All magmatic sulphides analysed have $\Delta^{33}S$ values within error of the terrestrial mass-dependent fractionation line (i.e. 0‰), averaging 0.01 ± 0.06‰ (Fig. 3). However, they exhibit a large $\delta^{34}S$ range from +1.0 ± 0.4‰ to −9.6 ± 0.4‰, and display a variety of morphologies and compositions (Figs 1, 3–5, Table 1).

The majority of sulphide inclusions analysed ($n = 33$) are homogeneous, smooth, circular to elliptical droplets (Fig. 1b) with compositions corresponding to monosulphide solid solution (mss) close to the pyrrhotite (Fe$_{1-x}$S) end-member (Fig. 4a, Supplementary Data 2). These sulphides only occur completely enclosed in phenocrysts, and have $\delta^{34}S$ values between +1.0 and −4.0‰ (Figs 3 and 5). We refer to them as group 1.

Another set of inclusions ($n = 12$), group 2, display heterogeneous mineralogy and more angular shapes (Fig. 1c–e). These consist of intergrowths of mss, intermediate solid solution (iss)[24–26] and an interstitial oxide phase. The mss in these inclusions reaches higher Ni (0.2–6.4 wt.%) and Cu (0.8–5.8 wt. %) contents than the homogeneous sulphides (0.1–2.2 wt.% Ni

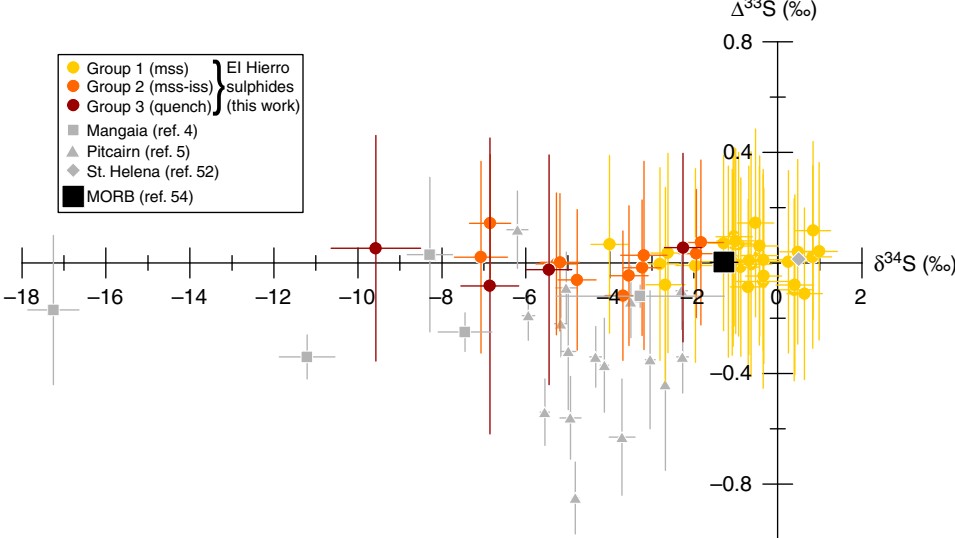

**Fig. 3** $\Delta^{33}S$ vs $\delta^{34}S$ for clinopyroxene- and spinel-hosted sulphide inclusions from El Hierro (coloured symbols, this study), compared to other OIB (grey symbols). The majority of inclusions have mss compositions close to pyrrhotite (yellow symbols, group 1), but some also contain Cu-rich iss intergrowths (orange symbols, group 2), while others display quench textures with low EPMA totals (~95%; dark red symbols). The $\delta^{34}S$ range is comparable to that of the silicate melt, with the most negative values reached by the 2-phase and heterogeneous sulphides. $\Delta^{33}S$ values cluster around 0‰ (i.e. within the mass-dependent fractionation line), contrasting with the Mangaia sulphides (grey squares; ref. [4]) and Pitcairn sulphides (grey triangles; ref. [5]) but similar to St. Helena glass[52]. The MORB field (black square; ref. [54]) is shown for comparison. The modelled effect of degassing (Fig. 2a) follows the negative $x$-direction at $\Delta^{33}S = 0$ (i.e. the terrestrial mass-dependent fractionation line). Error bars are $1\sigma$ propagated analytical uncertainties

## Discussion

Decompression-induced degassing beginning at pressures of at least 300 MPa was the primary cause of $CO_2$, $H_2O$ and S loss in the El Hierro magma[17] (see Supplementary Fig. 5). Sulphur speciation data ($S^{6+}/\Sigma S$) in the melt inclusions also show that the magma became more reduced during decompression and S loss to a fluid phase (Fig. 2b). However, the presence of sulphide globules in minerals also attests to S partitioning into an immiscible sulphide melt. The solubility of sulphur in magma before the separation of an immiscible sulphide liquid, also known as the sulphur concentration at sulphide saturation (SCSS)[32,33], increases steeply with increasing oxygen fugacity ($fO_2$) between the fayalite–magnetite–quartz (FMQ) buffer and two log units above it (FMQ + 2), corresponding to a switch from sulphide($S^{2-}$)- to sulphate($SO_4^{2-}$)-dominated conditions[33]. The most sulphur-rich melt inclusion has a S concentration (5080 ppm) well exceeding the S solubility of a relatively reduced magma (around FMQ)[32,33], but at FMQ + 1.5 dissolved S can reach >1 wt.%[33]. The $S^{6+}/\Sigma S$ ratio of that inclusion suggests a minimum $fO_2$ of 1.2 log units above FMQ (but likely higher; see ref. [17]), hence we infer that the initially oxidised magma was not sulphide saturated. The presence of sulphide droplets therefore reflects the decrease in $fO_2$ until intersection of the SCSS (Fig. 2b), at which point a sulphide liquid started to separate from the silicate melt. The absence of sulphide droplets in olivine phenocrysts supports a late saturation of sulphide. Alternatively, the lack of olivine-hosted sulphide inclusions could reflect preferential late-stage nucleation near rapidly-growing Fe–Ti oxide or clinopyroxene phenocrysts[17].

The crystallisation of magnetite ($Fe^{2+}Fe^{3+}{}_2O^{2-}{}_4$) has previously been argued to trigger sulphide saturation in hydrous, oxidised magmas from subduction zone settings[34–36]. This process could also reasonably be invoked for the El Hierro magma, supported by the ubiquitous association of sulphides with Fe–Ti oxides. However, the El Hierro oxides, with an average composition approaching $Fe^{2+}{}_{1.0}Mg^{2+}{}_{0.4}Fe^{3+}{}_{0.9}Al^{3+}{}_{0.3}Ti^{4+}{}_{0.4}O^{2-}{}_4$ (ref. [37]), have a ferric iron content less than half that of the magnetite end-member, and so their crystallisation may not have

modified the melt $Fe^{3+}/\Sigma Fe$ ratio sufficiently to trigger sulphide saturation[34]. Moreover, there is no systematic variation between the FeO content of melt inclusions and their $S^{6+}/\Sigma S$ ratios[17], which would be expected if spinel fractionation was the cause of magma reduction. Hence, we favour degassing-induced reduction[17,38–41] as the cause of the intersection of the SCSS and separation of an immiscible sulphide liquid. Our observations are consistent with different degassing modelling trends obtained with the software D-Compress[39], which all show reduction of the magma associated with S loss when the initial magma is oxidised (Fig. 2b). It is worth noting that while sulphide droplets are abundant in some phenocrysts, overall they represent a rather insignificant fraction of the S budget of the El Hierro magma (estimated at <1% of total S from thin section observations, see Methods), and therefore sulphide separation did not appreciably decrease the S content of the magma.

We now explore the intricate isotopic interplay among gas loss, sulphide saturation, and magmatic $fO_2$ with quantitative calculations of S isotope fractionation during magma ascent. The positive exponential correlation of S content and $\delta^{34}S$ in melt inclusions and matrix glasses (Fig. 2a) suggests that S loss via degassing is the primary process driving the strong mass-dependent isotopic fractionation, especially after a significant amount of S has escaped. There is no need to invoke mixing of isotopically distinct melts, which might be expected to produce linear arrays in Fig. 2a, and no evidence for late-stage assimilation of seawater-influenced components, which should drive $\delta^{34}S$ to high positive values[15]. Isotopic fractionation of sulphur in magmatic systems depends strongly on the proportions of oxidised and reduced S species in the different phases present, since oxidised species are generally enriched in $^{34}S$ relative to reduced species in the order $SO_4^{2-}{}_{(melt)} > SO_{2(gas)} > H_2S_{(gas)} \approx S^{2-}{}_{(melt)} \approx S^{2-}{}_{(FeS)}$ (ref. [15]). Since the proportions of oxidised and reduced S species in the melt are linked to the oxidation state of the magma, as monitored by $fO_2$ (ref. [33]), processes involving the exchange of sulphur between different phases (melt, gas or sulphide liquid) have the potential to induce isotopic fractionation when these

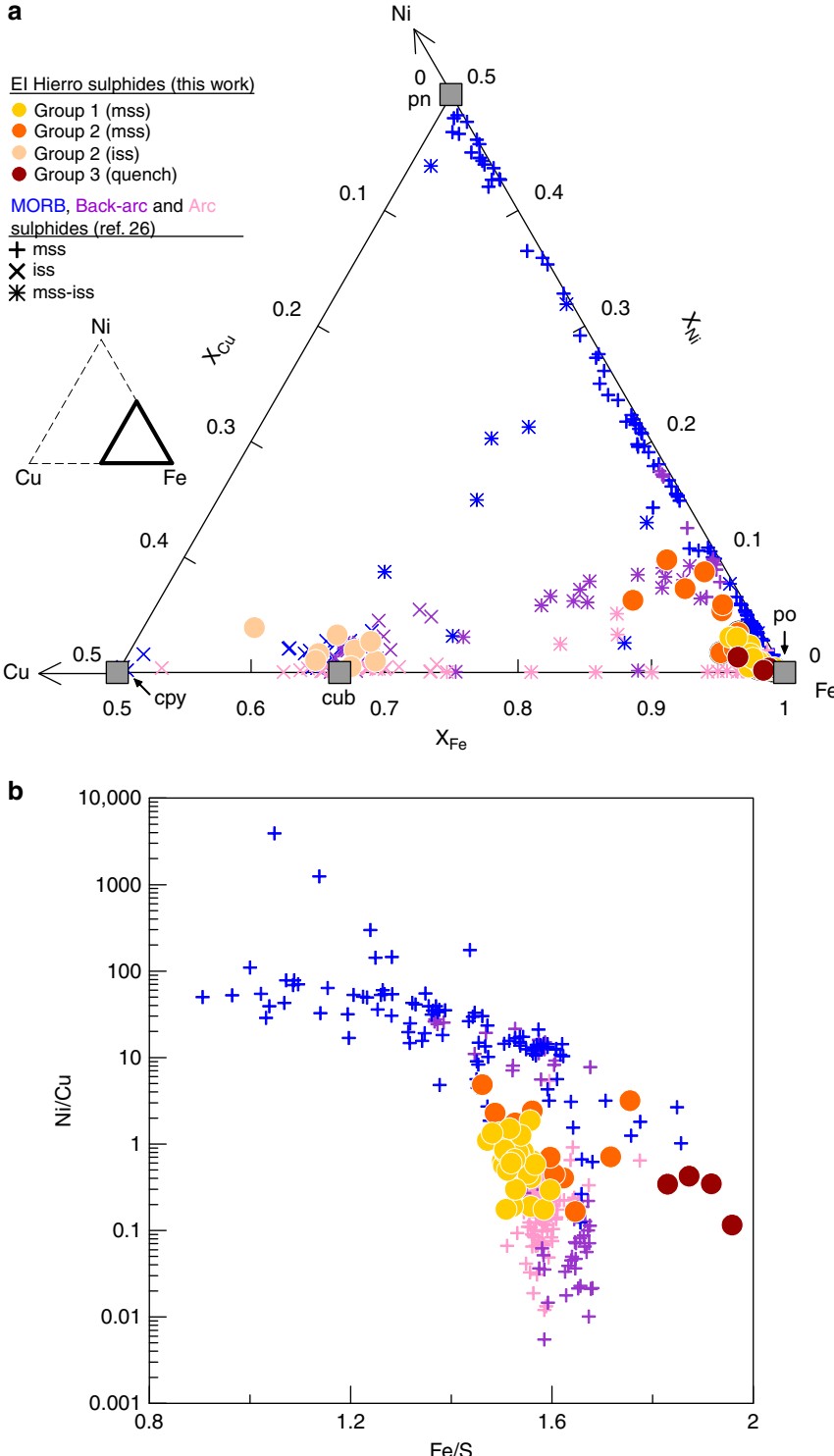

**Fig. 4** Geochemistry of El Hierro sulphides compared to sulphides from MORB, back-arc and arc settings. **a** Ternary diagram showing the composition (in molar fractions) of El Hierro sulphides (circles) and sulphides from MORB (blue symbols), back-arc (purple symbols) and arc settings (pink symbols)[26]. Different symbols are used for mss (crosses), iss (x's) and mss–iss intergrowths or zoned sulphides where phases were not separately measured (stars). El Hierro group 1 sulphides have pyrrhotite-like (po) mss compositions with increasing Ni content towards group 2. The mss and iss phases in group 2 sulphides were analysed separately (orange and beige circles, respectively), hence the bulk compositions of those globules would fall on a straight mixing line between the two. Common sulphide mineral compositions are also shown for comparison: pyrrhotite (po), pentlandite (pn), cubanite (cub) and chalcopyrite (cpy). Bold triangle in inset shows the region of the Fe–Ni–Cu ternary used for the plot. **b** Ni/Cu and Fe/S ratios (wt.%/wt.%) of mss as in ref. [26]., showing that the early group 1 sulphides have a strong affinity with arc sulphides and plot near the upper Ni/Cu end of the arc trend. The apparent extension of the MORB trend by group 3 sulphides is likely an artefact of their lower S content due to S degassing

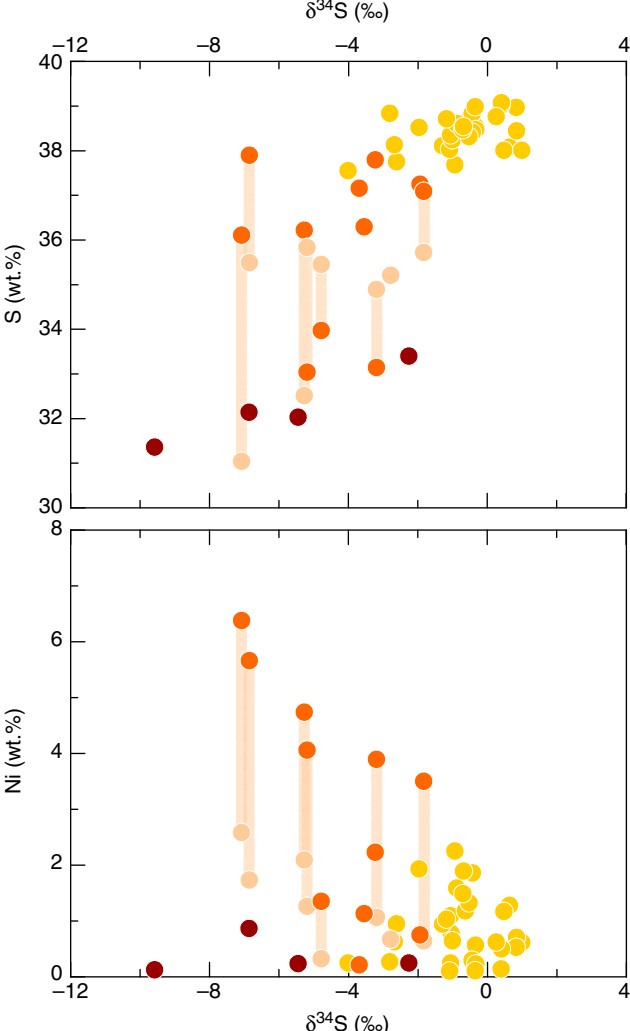

**Fig. 5** Variations of sulphide geochemistry with isotopic composition. Covariation of sulphur (**a**) and nickel (**b**) contents with isotopic composition, showing a general decrease in $\delta^{34}S$ with decreasing S content, and different trends for Ni which separate group 2 and 3 sulphides. Symbols as in Figs 3–4. Bulk compositions of group 2 sulphides plot along the pale orange bands, which join the S (**a**) and Ni (**b**) contents of the mss and iss intergrowths

exchanges involve modifications in the valence state of S. For example, the isotopic fractionation factor between a sulphide liquid and S dissolved in the melt ($\alpha_{FeS-melt}$, where $1000\ln\alpha_{FeS-melt} \cong \delta^{34}S_{FeS} - \delta^{34}S_{melt}$) will vary strongly with the proportions of dissolved $SO_4^{2-}$ and $S^{2-}$ in the melt (i.e., the $S^{6+}/\Sigma S$ ratio). At oxidised conditions (e.g. $\Delta FMQ \approx +1.5$), where significant $SO_4^{2-}$ is present, $\delta^{34}S_{FeS}$ is ~3‰ lower than $\delta^{34}S_{melt}$[15,42,43], such that sulphide segregation would cause the melt to become enriched in $^{34}S$. At reduced conditions, S has the same valence (−2) in both sulphide and silicate melts, and the isotopic fractionation is small, with $\delta^{34}S_{FeS}$ about 0.4‰ higher than $\delta^{34}S_{melt}$ at 1150 °C (ref. [15]). This fractionation and the small contribution of sulphide to the total initial magmatic S content are not sufficient to produce the strong isotopic signal we observe (Fig. 2a). Instead, the positive exponential trend between S content and $\delta^{34}S$ values in the melt implicates open-system degassing as the primary agent of isotopic change (Fig. 2a), even though gas loss is conventionally expected to favour light isotopes, a common misconception for redox-dependent isotope systems[18].

The sulphur species incorporated in a fluid phase at magmatic conditions are mainly $H_2S$ and $SO_2$ (refs [15,32]), in which S has a valence of −2 and +4, respectively. The proportions of $H_2S$ and $SO_2$ are therefore controlled by $fO_2$, but they also depend on $H_2O$ fugacity ($fH_2O$), which can be used as a proxy for pressure since $H_2O$ is the major constituent (in mol%) of the fluid phase (Supplementary Fig. 6). This can be described by the equilibrium reaction:[8,14]

$$H_2S_{(gas)} + 1.5O_{2(gas)} \rightleftharpoons SO_{2(gas)} + H_2O_{(gas)} \quad (1)$$

Degassing-induced S isotope fractionation is therefore sensitive to both $fO_2$ and $fH_2O$ (i.e. pressure)[14] (Fig. 6). As a result, both of these intensive parameters must be known simultaneously to correctly model degassing-induced isotopic fractionation of sulphur during the evolution of the El Hierro magmatic system. We used the $S^{6+}/\Sigma S$ and $CO_2$–$H_2O$ data of Longpré et al.[17] combined with the parameterisations of Jugo et al.[33] and Iacono-Marziano et al.[44] to estimate $fO_2$ and total pressure, respectively, in olivine-hosted melt inclusions (Fig. 2b and Supplementary Fig. 5). We considered two sets of end-member estimates[42,45] for the fractionation factors among the relevant S species ($SO_4^{2-}{}_{(melt)}$, $S^{2-}{}_{(melt)}$, $SO_{2(gas)}$, $H_2S_{(gas)}$)[15,32]. This allowed us to calculate a range of evolving gas–melt S isotope fractionation factors ($\alpha_{gas-melt}$) during gradual loss of sulphur from the El Hierro melt upon ascent from 300 to 1 MPa.

Under isothermal decompression at 1150° C[17], calculations based on our estimated $fO_2$ and pressure ranges produce lower- and upper-bound isotopic fractionation curves encompassing our entire dataset (Fig. 2a). The most recent experimentally determined fractionation factors[45] better reproduce the strongly fractionated $\delta^{34}S$ values we observe at low S abundances (Fig. 2a); details on these calculations are provided in the Methods section, and the differences between models are outlined in the Supplementary Discussion. Supplementary Data 3 contains the calculations used to model S isotope fractionation. Equivalent calculations, assuming S loss via sulphide segregation alone, demonstrate that this mechanism did not affect the S isotope fractionation we observe (Fig. 2a). As the $\delta^{34}S$ range of sulphide inclusions mirrors that of the melt inclusions and matrix glasses, we suggest that trapped sulphides essentially track the isotopic composition of the silicate melt as sulphur is degassed[15] (see below). Our calculations demonstrate that increased proportions of $SO_2$ ($S^{4+}$) in the gas phase, which is predicted at lower pressures, favour the degassing of heavy S isotopes since S in the El Hierro melt speciates toward $S^{2-}$ as oxygen fugacity decreases at lower pressures. While degassing of $SO_2$ had previously been identified as a potential trigger for magma reduction and isotopic fractionation[18,38,40,46], our data clearly demonstrate that it can exert the primary control on both in a positive feedback loop driven by decompression.

Compositional and textural trends in the magmatic sulphides can be explained as part of this framework. The smooth, homogeneous sulphides (group 1), which have the highest $\delta^{34}S$ values (Table 1, Figs 3 and 5), represent the early sulphide liquid that was entrapped as pristine mss droplets in growing crystals of clinopyroxene and spinel. Sulphides occurring at the edge of crystals and in contact with the matrix glass (groups 2 and 3) reach significantly more negative $\delta^{34}S$ values and exhibit more diverse features, such as angular shapes, heterogeneous mineralogy or quench textures. Moreover, these sulphides have a lower S content (Supplementary Data 2, Fig. 5a), suggesting that they formed or equilibrated in a S-poorer melt (i.e., at lower $fS_2$)[30,31], or were subject to S loss by degassing[27,47]. The Ni content of group 2 sulphides also increases in both mss and iss with decreasing $\delta^{34}S$ (Fig. 5b). To first order, the isotopic composition

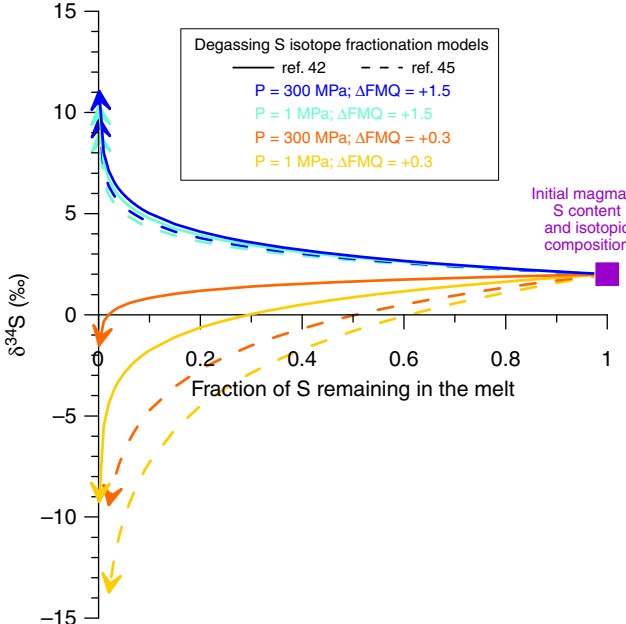

**Fig. 6** Effect of pressure and oxygen fugacity on S isotope fractionation. Degassing-induced S isotope fractionation modelling using the starting isotopic composition of the inferred parental El Hierro magma, showing the relative effects of pressure (P) and oxygen fucacity ($fO_2$). Strongly divergent paths are determined mainly by oxygen fugacity. Solid lines and dashed lines correspond to the fractionation factors of Miyoshi et al.[42] and Fiege et al.[45], respectively

of the sulphides should be controlled by the S isotopic composition of the dissolved sulphur species in the silicate melt from which they are segregating. The different groups thus likely represent different stages in the evolution of the sulphide melt, with the majority of droplets observed corresponding to the onset of sulphide saturation. A break in slope in the S vs FeO* trend of melt inclusions at a S concentration of ~1500 ppm may mark this onset[17]. This would have occurred after a significant amount of S was already degassed, but before the isotopic composition of the melt was substantially affected, consistent with the modest isotopic fractionation seen in the group 1 sulphides, with $\delta^{34}S$ between +1.0 and −2.8‰, with one exception reaching −4.0‰ (Fig. 2a, b).

The separation of a sulphide melt should also deplete the silicate melt in metals such as Ni and Cu, which have ~1000 times more affinity for sulphide liquid[25,48]. Therefore, the higher concentration of Ni and Cu in the group 2 sulphides cannot be taken as evidence for later saturation, since later-formed sulphides would precipitate from a more Ni- and Cu-depleted magma. Nevertheless, their occurrence at the outer edges and outside crystals combined with their $\delta^{34}S$ range of −1.8‰ to −7.1‰ suggests that they communicated with the silicate melt at a later stage than the group 1 sulphides, and may therefore have isotopically re-equilibrated with the $^{34}S$-depleted magma. The Ni and Cu partition coefficients between mss and sulphide liquid are <1 (ref. [28]), hence fractional crystallisation of mss should enrich the residual sulphide liquid in these elements. This scenario is consistent with the expected evolution of the sulphide melt, which should first crystallise mss, followed by iss at a later stage[49]. The group 2 sulphides therefore appear to represent residual sulphide liquid that crystallised iss and higher-Ni mss at a late stage while the $\delta^{34}S$ of the magma was plummeting, after removal of Ni-poorer mss (group 1).

The group 2 sulphides also crystallised an iron oxide phase along with mss and iss[24,50], implying that the residual sulphide liquid contained more oxygen, which is controlled by the reaction:[30,31]

$$FeS_{(sulphide\,melt)} + \frac{1}{2}O_{2(gas)} \rightleftharpoons FeO_{(sulphide\,melt)} + \frac{1}{2}S_{2(gas)} \qquad (2)$$

This is counterintuitive, considering that these later-crystallised sulphides formed at lower $fO_2$. However, incorporation of O in the sulphide liquid is also controlled by sulphur fugacity ($fS_2$)[30,31], where a lower $fS_2$ represents a competing effect to lower $fO_2$ and can result in a higher O content of the sulphide liquid. Model calculations[39] indicate that $fS_2$ decreased by several orders of magnitude more than $fO_2$ during degassing of the El Hierro magma (Supplementary Fig. 7), and thus exerted the dominant control on this equilibrium. The presence of O in the sulphide phase may also explain some of the lower EPMA totals observed in the group 2 and 3 sulphides (Supplementary Data 2).

The group 3 sulphides, in contrast, do not contain a discrete oxide phase and their Ni and Cu contents are low, but they also show a large $\delta^{34}S$ range towards very negative values. Their spherical shapes and fine trellis textures suggest quenching from sulphide liquid. We therefore interpret them as having formed at the latest stage from a Ni- and Cu-depleted silicate melt. Hence we propose that sulphide segregation occurred continuously once the magma reached sulphide saturation. The group 3 sulphides have the lowest S content, presumably because they did not crystallise oxide and thus retained O in the liquid. The lower S content of these sulphides should also have lowered their melting point[51], allowing them to remain a liquid phase, which is consistent with the observed quench textures. The overlap between the isotopic compositions of group 2 and 3 sulphides with group 1 (Table 1, Fig. 3) supports the idea of a continuous sulphide segregation history, with groups 2 and 3 experiencing larger fractionations at a later stage, as expected from the exponential decrease in $\delta^{34}S$ of the silicate melt upon S degassing (Fig. 2a).

The chemistry of mss sulphides has been shown to vary systematically between MORB and arc magmas, due to their different crystallisation histories and redox states[26]. The Ni content of sulphides is generally controlled by the amount of olivine fractionation that has occurred prior to sulphide saturation, since Ni is compatible in olivine. MORB sulphides have significantly higher Ni than sulphides from arc magmas (Fig. 4, Supplementary Fig. 8), which can be attributed to the lower $fO_2$ of MORB parental magmas and, hence, earlier sulphide saturation prior to extensive removal of olivine[26]. In contrast, arc magmas are generally more oxidised and reach sulphide saturation at a later stage, when the magma is more depleted in Ni. The chemistry of the El Hierro sulphides is most similar to sulphides from arc and back-arc settings, with lower Ni and Ni/Cu ratios (Fig. 4, Supplementary Fig. 8), which is consistent with the relatively high oxidation state of the inferred parental magma, and thus later sulphide saturation. These results are supported by the S isotope data, and also indicate that S degassing can have an additional important effect on the composition of the sulphides and stabilisation of the sulphide melt[47].

While the HIMU mantle reservoir is characterised by extreme $^{206}Pb/^{204}Pb$ ratios that constrain its age to >1 Ga to account for sufficient uranium decay[10,11], we find no S-MIF in El Hierro sulphides (Fig. 3), which is consistent with a post-Archaean (<2.5 Ga) origin for any recycled material in the Canary Island plume. Sulphur isotope analyses on basalt glasses from St-Helena, another HIMU-type hotspot in the South Atlantic, also show no

signs of MIF[52], contrasting with the MIF-bearing Mangaia sulphides presumably derived from an older recycled source[4]. It appears that HIMU-type signatures may not originate from a single reservoir, but rather from a recurrent geological process. This is consistent with the slightly lower $^{206}Pb/^{204}Pb$ ratios of the HIMU component found in the Canary Islands and other OIB from the eastern Atlantic Ocean, such as the Cape Verde islands and Madeira[53], which also suggests a shorter time-integrated addition of radiogenic Pb relative to Mangaia, and thus a more recent age of formation. By analogy, the Archaean MIF signal observed at Pitcairn[5] may not require that all EM-I OIB have Archaean plume sources. Moreover, while degassing cannot explain the negative MIF $\Delta^{33}S$ values observed at Mangaia[4] and Pitcairn[5], our results clearly show that it can impart negative $\delta^{34}S$ signatures to volcanic rocks and their S-rich inclusions, potentially overprinting the isotopic composition of the source. Degassing thus offers an alternative mechanism to produce the highly negative $\delta^{34}S$ values of the Mangaia and Pitcairn sulphides, which have been interpreted to reflect an intrinsic feature of the presumed Archaean protolith within these plumes[4,5,13]. Disentangling the effects of degassing from source heterogeneity is thus critical for understanding the geodynamic implications of S isotopes at volcanic hotspots.

By removing the effect of degassing, we find that the mantle source of the El Hierro magma was significantly enriched in $^{34}S$ compared to chondrites and the bulk silicate Earth[54], reaching a $\delta^{34}S$ value of $+3.2 \pm 0.1‰$ in S-rich melt inclusions. A strikingly similar value of $+3.5‰$ was previously reported for S-rich clinopyroxene-hosted melt inclusions from Gran Canaria where, like at El Hierro, the most S-rich inclusions also show the highest $S^{6+}/\Sigma S$ ratios[6]. The convergence of these results suggests a potential link between high oxidation state and enriched $\delta^{34}S$ in the mantle sources of volcanic hotspots. This connects two emerging views: (1) in line with other isotopic proxies[11,21,53], recycling of subducted sulphur best explains S isotope signatures observed at many OIB hotspots (Mangaia[4], Pitcairn[5], Samoa[9], and now the Canary Islands), and (2) the mantle sources of OIB appear to be more oxidised than those of MORB[33,41], but less so than those of arc magmas[46,55]. During subduction, altered and oxidised oceanic crust and sediments transfer redox potential to the mantle wedge underlying volcanic arcs, likely with sulphur as an oxidising agent[46,55]. However, a significant fraction of S might remain in the downgoing slab since estimates of S output at arcs are an order of magnitude lower than estimates of subducted S (refs [46,56].), which may act as a potent provider of S and oxidising power in hotspot sources. The source of subducted S can be generally constrained from the S isotope compositions inferred here. While sulphides produced by microbial sulphate reduction during alteration of the modern-day oceanic crust generally show negative $\delta^{34}S$ values[57,58], this may have been different at earlier stages in Earth's history[1]. The high $\delta^{34}S$ ($+2.8‰$) of recycled S at Samoa, for example, has been attributed to the positive $\delta^{34}S$ that characterizes sulphides in Proterozoic sediments[9]. On the other hand, serpentinization of oceanic peridotites may be an important sink for the S in high-$\delta^{34}S$ seawater sulphate[59], as recent evidence points toward significant fluid uptake during bending and faulting of the oceanic lithosphere before subduction[60,61]. Whether the subducted slab remains more oxidising than the surrounding mantle at greater depths and over long timescales is, to our knowledge, poorly constrained, but redox heterogeneities introduced into the deep mantle via subduction[46,55,62] conceivably play an important role in global geochemical dynamics, as documented by diamond-forming processes[62,63]. The survival of such heterogeneities until recycling at hotspots could be behind the documented S isotope differences between OIB and MORB, and may explain their apparent redox contrast as well.

## Methods

**Sample collection and preparation.** Lava balloons floating on the surface of the ocean were collected during the eruptive activity at El Hierro on 27 November 2011, 6 December 2011 and 28 January 2012. These samples were crushed and sieved, and phenocrysts of clinopyroxene and spinel were picked and mounted in epoxy to expose sulphide inclusions. Silicate melt inclusions and matrix glasses were previously mounted in indium and analysed for volatile abundances and sulphur speciation[17].

**Secondary ion mass spectrometry.** In situ sulphur isotope measurements on these same glasses ($^{32}S$, $^{34}S$) and on the sulphide inclusions ($^{32}S$, $^{33}S$, $^{34}S$) were performed using a CAMECA 1270 SIMS instrument at the University of California, Los Angeles (UCLA). A $Cs^+$ primary beam was focused to a 10–15 μm spot size at a mass resolving power of 5500, with $^{32}SH^-$ fully resolved from $^{33}S^-$ for the triple isotope measurements.

The sulphides were measured in multi-collection mode with three Faraday cups, with beam current set at 5–6 nA. Charge buildup on the gold-coated sample surface was compensated using an electron flood gun. A cold finger surrounding the sample connected to a Dewar flask of liquid $N_2$ was used to reduce hydrides in the sample chamber. A contrast aperture and field aperture were used to exclude aberrant ions. Beam centering in the Field Aperture was scanned before each analysis. Ions from an energy window of 30 V were collected. The magnetic field was set and subsequently controlled by Nuclear Magnetic Resonance. Isotope counts per second (CPS) were measured in multi-collection mode with three Faraday cups. Slits of a common size on each detector, together with entrance slits, contrast aperture and energy window, yielded a mass resolution power of ~ 5500 (mass/$\Delta$mass) with $^{32}SH^-$ fully resolved from $^{33}S^-$. Faraday cup electrometers were calibrated immediately prior to analysis.

The analytical routine included a 60 s pre-sputter to clean the sample surface, centre the beam in the field aperture and achieve sputtering equilibrium before data collection. Measurements of each isotope in counts per second were collected in 6 cycles of 10 sec each to allow time for settling of the Faraday cup electronics between cycles. Standards used to calibrate the instrumental mass fractionation (Supplementary Table 1) included Balmat (pyrite), CAR123 (pyrite), Canyon Diablo Troilite and Anderson pyrrhotite[64,65], which was also used as our in-run internal standard. The resulting calibration curve is shown in Supplementary Figure 1 and has a slope of $0.517 \pm 0.002$ ($2\sigma$). Triple sulphur isotope measurements ($n = 55$; Supplementary Table 2) ($^{32}S$, $^{33}S$ and $^{34}S$) were then obtained on exposed sulphide inclusions ranging in size from 30 to 100 μm. Regularly interspersed measurements (every 5–6 measurements) of our reference material, Anderson pyrrhotite ($n = 15$), gave a standard deviation ($1\sigma$) on $\delta^{34}S$ of 0.4‰, compared with its known $\delta^{34}S$ of $1.4 \pm 0.3‰$[64].

Double isotope analyses ($n = 44$; Supplementary Data 1) were conducted on glasses using instrumental parameters and analytical technique identical as above except for the following differences: (1) $^{34}S$ was measured on an electron multiplier; (2) the primary beam current was reduced to 3.15 nA in order to balance the errors associated with high count rates on the electron multiplier (deadtime error) and low count rates on the Faraday cup (background); (3) analyses consisted of 20 cycles of 10 s each. The standard used for calibration is P1326-2, described below.

Three repeat analyses of one large sulphide inclusion (EH_4-29), done at different times during the analytical session, show good reproducibility, yielding $\delta^{34}S$ and $\Delta^{33}S$ values of $-0.64 \pm 0.06‰$ ($1\sigma$) and $0.00 \pm 0.06‰$ ($1\sigma$), respectively. Other sulphide inclusions analysed twice ($n = 5$) are reproducible within error, except for the most isotopically negative inclusion ($\delta^{34}S = -8.9 \pm 0.4‰$ and $-10.2 \pm 0.4‰$).

Matrix glasses show more heterogeneity; for example, three analyses of a single glass chip (# 9.25) give a $\delta^{34}S$ range of 2.5‰, with an average of $-0.9 \pm 1.3$ ($1\sigma$). Other glass chips analysed twice ($n = 6$) show as much as 3.8‰ difference between two measurements (see Methods for discussion of uncertainty).

**Description of glass standard.** Our glass standard used in SIMS analyses, P1326-2, is a MORB glass from the Juan de Fuca ridge, and has previously been described for major and trace elements by refs [66,67]. It is highly homogeneous and contains very few microlites. Its volatile concentrations ($CO_2$, $H_2O$, F, S, Cl) have previously been measured by SIMS, on separate occasions by refs [17,67], which showed good reproducibility. Ref. [17] reports $340 \pm 24$ ppm $CO_2$, $0.29 \pm 0.09$ wt.% $H_2O$, $206 \pm 6$ ppm F, $1296 \pm 26$ ppm S and $187 \pm 5$ ppm Cl.

**Isotope ratio mass spectrometry (IRMS).** The sulphur isotopic composition of P1326-2 was independently analysed by both $SO_2$ continuous flow at the University of New Mexico (Thermo Delta Plus XL mass spectrometer) and $SF_6$ dual inlet isotope-ratio mass spectrometry at McGill University (Thermo Scientific MAT 253). The $SO_2$ measurements were converted to the $SF_6$ scale using the regression of ref. [68]. Sulphur extractions were achieved in two ways: (1) the Kiba reagent extraction method, which extracts all S species, and (2) combined chromium reduction and Thode reagent methods to separately extract $S^{2-}$ and $S^{6+}$, respectively[69,70,71]. The extracted sulphur was re-precipitated as silver sulphide ($Ag_2S$), before fluorination to $SF_6$ and subsequent isotopic analysis. For the separate sulphide and sulphate extractions, we used the relative weights of $Ag_2S$ precipitates

to estimate the $S^{6+}/\Sigma S$ ratio, followed by a mass balance to obtain a bulk $\delta^{34}S$ value. Replicate extraction and analysis ($n = 3$) give $\delta^{34}S = 0.8 \pm 0.1$‰ and $S^{6+}/\Sigma S = 0.20 \pm 0.03$. The full results of these analyses are summarised in Supplementary Data 4.

**Electron probe micro-analysis (EPMA).** The major and minor element compositions of sulphides were determined at McGill University on a JEOL 8900 electron microprobe, using an acceleration voltage of 20 kV, a beam current of 30 nA and a beam size of 3 μm. The instrument was calibrated with the following standards: AsCoNi alloy (for As, Co and Ni), pyrrhotite and chalcopyrite (for Fe, S and Cu), galena (Pb) and sphalerite (Zn). A total of 55 analyses were obtained and are presented in Supplementary Data 2, including one analysis of Anderson pyrrhotite, which agrees very well with known Anderson pyrrhotite composition ($Fe_{0.87}S$)[64].

**Scanning electron microscopy (SEM).** Elemental maps were acquired for six different sulphide inclusions (Supplementary Figs 3–4) using a Hitachi SEM at Queens College, City University of New York, equipped with a Bruker Quantax 400 energy-dispersive X-ray spectroscopy (EDS) detector.

**Estimation of the S budget of magmatic sulphides.** Thin section observations show that sulphides, while abundant in some clinopyroxene and spinel phenocrysts, actually represent a minor component of the El Hierro magma. A conservative estimate for the mass fraction of sulphide in the bulk rock is 0.01 wt.%. Assuming an FeS composition, the sulphides would then have ~36 wt.% S. This corresponds to an upper limit for the sulphide contribution to bulk S content of 36 ppm, hence <1% of the total initial S concentration (5080 ppm).

**Calculation of uncertainty.** The instrumental mass bias factor $\alpha_i$ on the isotopic ratio $^{34}S/^{32}S$ was calculated by comparing the average of the raw measured ratios ($R_{avg}$) of an internal calibration standard with its known isotopic ratio:

$$a_i = R_{avg}/R_{known} \tag{3}$$

For the sulphide inclusions the standard was Anderson pyrrhotite ($R_{known} = {}^{34}S/{}^{32}S_{known} = 0.0442244$ or $\delta^{34}S_{V\text{-}CDT} = +1.4 \pm 0.3$‰)[64]. The calculation of $\alpha_i$ for $^{33}S/^{32}S$ was done similarly, assuming a $\delta^{33}S_{known}$ falling on the regression line between $\delta^{33}S$ and $\delta^{34}S$ shown in Supplementary Fig. 1, i.e. dependent on $\delta^{34}S_{known}$. The uncertainty on $\alpha_i$ is given by:

$$\sigma_\alpha = \sqrt{\left(\frac{\sigma_{avg}}{R_{known}}\right)^2 + \left(-\sigma_{known}\frac{R_{avg}}{R_{known}^2}\right)^2} \tag{4}$$

where $\sigma_{avg}$ is the standard deviation on the average of the raw measured ratios and $\sigma_{known}$ the uncertainty on the known isotopic ratio of the standard.

The measured ratios $^xS/^{32}S$ ($R_i$) were then divided by $\alpha_i$ to obtain the corrected, 'true' isotopic ratios, and then converted to $\delta^xS_{V\text{-}CDT}$ values:

$$\delta^xS_{V\text{-}CDT} = \frac{R_i}{R_{V\text{-}CDT}} - 1 \tag{5}$$

where $R_{V\text{-}CDT}$ is the isotopic ratio of the international reference standard Vienna-Canyon Diablo Troilite[64]. The propagated analytical error on $\delta^xS$ values is then calculated with the following equation:

$$\sigma_{\delta^xS} = \left(\frac{1}{\alpha \times R_{V\text{-}CDT}}\right) \times \sqrt{\sigma_i^2 + \left(\frac{-R_i}{\alpha}\sigma_\alpha\right)^2} \tag{6}$$

where $\sigma_i$ is the analytical uncertainty on the raw isotopic ratios ($R_i$).

Once the corrected isotopic ratios are converted to $\delta^{34}S_{V\text{-}CDT}$ and $\delta^{33}S_{V\text{-}CDT}$ values, $\Delta^{33}S$ is calculated with the following expression:

$$\Delta^{33}S = \delta^{33}S_{V\text{-}CDT} - [(\delta^{34}S_{V\text{-}CDT} + 1)^k - 1] \tag{7}$$

where $k = 0.5166$ represents the slope of the mass-dependent fractionation line in $\delta^{33}S_{V\text{-}CDT}$ - $\delta^{34}S_{V\text{-}CDT}$ space (Supplementary Fig. 1). The analytical uncertainty on $\Delta^{33}S$ then becomes:

$$\sigma_{\Delta^{33}S} = \sqrt{\sigma_{\delta^{33}S}^2 + \sigma_{\delta^{33}S}^2 \times \left[k \times \left(1 + \delta^{34}S\right)^{-(1-k)}\right]^2 + 2\sigma_{\delta^{33}S\delta^{34}S} \times \left[k \times \left(1 + \delta^{34}S\right)^{-(1-k)}\right]} \tag{8}$$

Duplicates were measured on 6 sulphide inclusions, and always agreed within uncertainty ($1\sigma$), except for the most $^{34}S$ depleted inclusion, which had individual $\delta^{34}S$ values of $-8.9 \pm 0.4$‰ and $-10.2 \pm 0.4$‰ (average of $-9.6 \pm 1.1$‰).

For the sulphide inclusions, no instrumental drift was observed between the different standard measurements during the analytical session. For the melt inclusions and matrix glasses, analyses were conducted in a 2-day span, with 31 analyses on the first day and 13 on the second day. A linear correction was applied

to the first day measurements to account for instrumental drift observed on measurements of our standard, P1326-2 ($n = 24$) (Supplementary Fig. 2). On the second day, no instrumental drift was observed.

The sulphur isotope data on glass measurements were corrected for instrumental mass bias ($\alpha_i$) using the average of measured $^{34}S/^{32}S$ ratios on P1326-2 (day 1: $n = 24$, $\alpha_i = R_i/R_{known} = 0.99223$; day 2: $n = 13$, $\alpha_i = 0.98871$) compared to our independently determined value for P1326-2 ($R_{known} = {}^{34}S/{}^{32}S_{known} = 0.04420$ or $\delta^{34}S_{V\text{-}CDT} = 0.9 \pm 0.3$‰).

Errors on $\delta^{34}S$ values were computed as for the sulphide inclusions and range from 0.1 to 0.8‰.

**Modelling of S isotope fractionation.** The positive correlation between $fO_2$ (in $\Delta FMQ$ log units, derived from $S^{6+}/\Sigma S$ ratios) of melt inclusions and their S content is used to estimate $fO_2$ variation during degassing (Fig. 2b), but as mentioned in the main text the highest $S^{6+}/\Sigma S$ values obtained are minimum estimates, and their variability in the matrix glasses is high. To account for this uncertainty, our approach has been to place maximum and minimum bounds on $\Delta FMQ$ values based on the log $fO_2$ ($\Delta FMQ$) vs. S trend shown in Fig. 2b. This results in a sensitivity analysis both minimising and maximising the extent of S-isotope fractionation.

Vapour saturation pressures based on volatile concentrations in melt inclusions were calculated using the data of Longpré et al.[17] and the model of Iacono-Marziano et al.[44]. The relationship between S content and the calculated pressures for olivine-hosted melt inclusions is shown in Supplementary Fig. 5. Spinel-hosted melt inclusions were excluded because their $CO_2$ contents are thought to be affected by disequilibrium degassing and overestimate pressures significantly[72]. Matrix glasses were also excluded because they are supersaturated with respect to $CO_2$ (ref. [17]), giving calculated pressures significantly in excess of the hydrostatic pressure corresponding to eruption depth. A polynomial fit was applied to the data in order to obtain input pressures as a function of S content in the degassing isotopic fractionation model. Five data points from the January 2012 sample were excluded from the computation of the polynomial fit, as they exhibit a slightly different degassing behaviour, characterised by lower $H_2O$ and S content for comparatively high $CO_2$.

These trends can be compared to degassing paths obtained with the software D-Compress[39]. This model requires the initial pressure, $fO_2$, $CO_2$ and $H_2O$ contents of the magma as inputs, which we estimate from the most volatile-rich melt inclusion. We start with an initial $CO_2$ content of 3420 ppm and $H_2O$ content of 3.05 wt.%, pressures varying between 300 and 400 MPa and $fO_2$ conditions between $\Delta FMQ = 1.2$ and 1.7 (ref. [17]). The programme outputs the initial S content and models each of these parameters ($CO_2$–$H_2O$–$S$–$fO_2$) during decompression. While the output from D-Compress generally provides good fits to the melt inclusion data (Fig. 2b, Supplementary Fig. 5), we note two main discrepancies: (1) the output S content reaches minimum values of >1100 ppm at low pressure (~1MPa), more than twice higher than what we measure in the matrix glasses (~500 ppm); (2) computed melt $S^{6+}/\Sigma S$ ratios stay relatively constant as $fO_2$ decreases. This decoupling of $S^{6+}/\Sigma S$ and $fO_2$ in D-Compress may arise because the $S^{6+}/\Sigma S$ vs. $fO_2$ relationship given by Jugo et al.[33], used in our conversion of $S^{6+}/\Sigma S$ ratios to $\Delta FMQ$ values, could not be implemented in D-Compress (A. Burgisser, personal communication). Nevertheless, there is general agreement that $fO_2$ decreases with S and $H_2O$ degassing. This is also supported by XANES data on $Fe^{3+}/\Sigma Fe$ ratios in El Hierro melt inclusions (Y. Moussallam, personal communication).

**Code availability.** An annotated version of the spreadsheet used to construct the isotopic fractionation trends in Fig. 2b is included as Supplementary Data 3.

## Data availability
All data generated during this study are included in the published article and Supplementary Data files.

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

## Acknowledgements

We thank the Spanish Instituto Geográfico Nacional for providing lava balloon samples from the submarine El Hierro eruption. This work was completed as part of P.B.'s Master's thesis at CUNY Queens College, supported by awards from the Fonds de recherche Nature et technologies du Québec (FRQNT), the Geological Society of America and the Association of Environmental and Engineering Geologists to P.B., a Queens College Research Foundation grant, including partial support from the Paula and Jeffrey Gural Endowed Professorship in Geology, to M.-A.L. and NSERC Discovery and Accelerator grants to J.S. The Stable Isotope Laboratory at McGill University was supported by the FRQNT through the GEOTOP research centre. We thank Lang Shi for technical assistance with electron microprobe analyses of sulphide inclusions and M. de Moor and M. Campbell for preliminary S isotope analyses of the P1326-2 glass standard. This work benefited from fruitful discussions with D. Baker, P. Cartigny, T.L. Grove, S. Ono and W. Blanford, comments by J. Marsh and N.G. Hemming, and formal reviews by M. Keith.

## Author contributions

M.-A.L. and J.S. conceived the project. P.B., M.-A.L. and R.E. performed the in situ SIMS analyses and P.B. conducted the EPMA work. P.B. and T.H.B. performed the S isotope analyses on the glass standards, with the help of B.W.; P.B. and B.W. conducted the calculations of degassing-induced sulphur isotope fractionation. All authors participated in the interpretation of the results. The paper was primarily written by P.B., with input from all authors.

## Additional information

**Competing interests:** The authors declare no competing interests.

