## [Peer Review File · Nature Communications]

Reviewers' comments:

Reviewer #1 (Remarks to the Author):

This paper uses the contents and isotope compositions of sulfur in melt inclusions and sulfide minerals to investigate correlations of sulfur with geochemical heterogeneities of the mantle. The authors show that degassing significantly affects the S isotope composition of the melts, but they can nevertheless obtain the primary S isotope composition of the mantle. They show that the El Hierro magma was enriched in ^{34}S compared to MORB mantle, and that there are no mass-independent fractionation effects for sulfur, indicating young enrichment relative to effects from recycled Archean rocks. The ^{34}S enrichment is similar to that observed in previous work for Samoa, and the authors contribute this to recycled sulfur. They suggest that low $\delta^{34}\text{S}$ values of recycled Archean sulfur in the Cook Islands may result from degassing.

These are new results and a valuable contribution to understanding recycling of volatiles and the relation of recycling of sulfur to other isotopes and trace elements that define the mantle end-members. The authors show that degassing effects can be significant and must be taken into account in evaluation of mantle source compositions.

The paper is well written, concise, and clear. The figures are clear and necessary, and the supplemental material is an important addition to the paper.

I only have a couple of questions.

Lines 109-111: it is suggested that the sulfide minerals track the isotope composition of the melt during degassing, but does your data show this? (e.g., Table S1?)

138-139: mantle heterogeneities in S isotopes are attributed to "recycling of subducted sulfur," but what is the source or reservoir of this sulfur? Altered oceanic lavas have negative $\delta^{34}\text{S}$ values (mean = -9‰ Rouxel et al 2008; Alt and Shanks, 2011), so is this sulfate in limestones or sediment pore waters? What about sedimentary sulfide? It seems that some mention of what this sulfur is recycled from is worthwhile here (or refer to a more detailed evaluation of subducted sulfur).

Reviewer #2 (Remarks to the Author):

1) Summary

The geochemistry of OIBs suggest that recycled subducted material modifies the hotspot mantle source. The authors use the S isotope system to better understand the recycling of S in the deep Earth, i.e. the source region of hotspots. Mass dependent ($\delta^{34}\text{S}$) and independent ($\delta^{33}\text{S}$) S isotopes have been used to investigate the effect of redox, S speciation and degassing on the isotope fractionation to define the S isotope composition of the hotspot source. For this purpose the authors also combined S isotopes with previously published Pb isotope data, which is diagnostic for a HIMU-type mantle source.

The structure of the manuscript suggests that it has previously been submitted to Nature or Nature Geoscience. However, the structure of the manuscript is also its major weakness and it would highly benefit of a clearer structure with sub-sections. For example, the methods are briefly described in the results (lines: 55-60, 73-75). I would recommend to present the methods in an own chapter. Similarly, the last paragraph of the results is interpretation (lines: 75-85) and should therefore be moved to the discussion.

Furthermore, the manuscript needs a more detailed description of the occurrence and chemistry of magmatic sulphides. I wonder (1) when the magmatic sulphides formed during melt evolution also with respect to degassing, (2) whether sulphide segregation in this system is a single, multistage or continuous process and (3) the most important question does this have any effect on their S isotope composition? Question (1) and (2) have been discussed in recent studies from Patten (2012 CanMineral; 2013 ChemGeol) and in our new study (Keith et al., 2017 ChemGeol).

This would significantly improve the manuscript and helps to support and highlight the interesting and important results and conclusions of this study. Hence, in my opinion the manuscript is generally suitable for publication in Nature Communications after major revisions. Please see my detailed comments as listed below.

2) Title

The study uses mass-dependent and -independent S isotopes, which is not clear based on the manuscript title. I think it would be important to highlight in the title that mass-dependent and -independent S isotopes are used, this could attract more readers.

3) Abstract

The abstract reflects the content of the article.

Lines 18-19: I don't really like the phrase "magma redox, S speciation and degassing". I recommend to change it to: "redox conditions, S specification and degassing of hotspot magmas."

Lines 20-21: This last sentence is not necessary. The previous sentence (lines: 18-19) represents a good conclusion of the study. Based on this last sentence the reader may wonder, why the authors talk about redox, speciation and degassing before but then reduce it to a degassing effect. Does this mean that only degassing is important? I would delete that sentence.

4) Introduction

The introduction lacks any background information about melt inclusions and magmatic sulphides, however, the manuscript focuses on their S isotope composition. A short overview of the recent literature must be added.

Line 23: Are these really all geochemical processes and no physical processes? I would rather write: "Chemical and physical processes within the mantle...".

Line 27: Change "mantle circulation" to "mantle convection".

Line 28: Based on the title the reader thinks that this paper is about mass-dependent fractionation (MDF) but here you talk about MIF. This has to be clarified. I would change the title as mentioned above and in addition in this part of the introduction you should tell the reader briefly what the difference between MDF and MIF is and what it can or cannot tell you based on previous studies and with respect to the aim of this study.

Line 28-31: I would not present such an equation in the introduction this should be part of the methods. Just write here that MIF is defined by the $\Delta^{33}\text{S}$ values. If necessary the authors can refer to the methods section for further details.

Line 32: Be consistent by using "S" or "sulphur". I would use "S" throughout the manuscript. I know what you mean but how do you get the Archean surface S into the mantle? I assume this is due to plate subduction.

Line 45-53: This paragraph should not be part of the introduction. Here you present the geological overview, as well as details which are commonly part of the petrographic and chemical results. Move this into different chapters/sections. Moreover, figures should not be presented in the

introduction, these are your results. The introduction has to provide an overview of previous research with respect to the aim of the study.

5) Results

Add a brief geological overview to the manuscript, which sets your samples into a geological contexts. If there are significant differences between different samples it would worth to produce a table to show this. This would help you to keep this chapter short.

The manuscript really lacks a petrographic section to present the results based on Fig. 1.

Here the authors should describe the host phases of the melt inclusions and magmatic sulphides. I wonder whether the magamtic sulphides are hosted only in one or in different phases. This is important because in the latter case the data would represent different stages of magma evolution. Furthermore, the authors should estimate the size of the melt and sulphide inclusions. This is important with respect to the beam diameter of the SIMS, please see also my comment below.

Lines 55-60: This should be part of the methods chapter, where the authors then can also define $\Delta^{33}\text{S}$ based on the equation presented in the introduction (line: 30). It makes sense to have some of the methods in the electronic supplement, however, the reader must understand and have the chance to evaluate the data without the supplement. Hence, I highly recommend to add a methods chapter. In addition, in this chapter the authors should tell the reader the general set up of the SIMS, i.e. what beam diameter has been used. This is important with respect to the size of the melt and magamtic sulphide inclusions. I wonder whether the signal was homogeneous or whether it sometimes represented a mixture of the inclusions and the host phase. This should briefly be described in the methods more details can then be shown in the supplement.

Line 60: Delete "quasi-".

Line 61: "...decline with S content" this is shown in Fig. 2a, so the authors should already refer to this figure here.

Lines 66-68: The authors refer here to Fig. 3 but I can't see the terrestrial mass dependent fractionation line in this diagram. It would be good to show it in the diagram or at least note it in the figure caption.

Line 68-69: The authors should refer to Fig. 2b again, where a homogenous inclusions is shown. Again, this is something which should be described in a petrographic results. According to Fig. 3 (supplement) I wonder why Ni hasn't been mapped because in Fig. 3a (supplement) the authors point out that Ni-rich sulphides occur, what I would expect in a basaltic melt. For the classification of the sulphides I would recommend to produce a triangular plot Fe-Ni-Cu including the stoichiometric compositions of pyrrhotite, pentlandite, cubanite and chalcopyrite. As for example, shown in Patten et al. (2012 CanMineral) and Keith et al. (2017, ChemGeol). This diagram may be part of the supplement.

Line 70-72: This is interesting so $\delta^{34}\text{S}$ varies between different types of sulphide inclusions. Hence, it is important to figure out when they formed during silicate melt evolution and whether they represent the same evolutionary stage.

Line 72: I disagree, Cu is in the same liquid as Fe, Cu and S when the silicate melt reaches the S saturation limit and an immiscible sulphide liquid forms. From this immiscible sulphide liquid then first mss (Fe-Ni-rich) exsolve followed by iss (Cu-Fe-rich). Please see Barnes et al. (2006, ContribMineralPetrol) and Patten et al. (2012, CanMineral). This is basically what I mean with presenting some recent results of this area of research in the introduction.

Lines 75-85: This is interpretation and has to be moved to the discussion. This part could be used as some introductory section for the discussion.

Line 77-78: This is exactly what I mean, so timing of sulphide segregation from the silicate melt affects $\delta^{34}\text{S}$. Hence, it is important to distinguish between earlier and later sulphides. That would be a really important result.

Lines 78-80: That's not clear to me, please clarify. Why "however" don't you write above that this is case? Maybe that's me but I struggle here.

Lines 82-84: Exactly, redox (and other parameters) control the S solubility in a silicate melt, and hence sulphide segregation can be a multistage process.

8) Discussion

I recommend to add a few sub-chapters/sections to the discussion. One of them could be introductory based on the last paragraph of the results, which I recommended to move to the discussion. I would also add a section discussing the effect of sulphide segregation at different stages during silicate melt evolution and its effect on the S isotope composition of the sulphides, if this is something that can be shown and discussed.

Line 90: I would replace "...vary in parallel with the oxidation state (as monitored by oxygen fugacity or fO_2)" by "... are representative for the oxidation state of the magma, as monitored by oxygen fugacity (fO_2).

Line 95: Please be consistent in using supplement or extended data. In Fig. 4 (supplement) please point out that the total pressure equals the fH_2O . I'm not sure whether this correct (please check) but this is what the authors write above (lines 91-92).

Line 98: Please be consistent in using fH_2O or total pressure, or is it not the same?

Line 99: What do you mean with "parameterization"? Conditions or set-up?

Lines 109-111: This could work but you have to know when degassing and sulphide segregation occur during melt evolution at least in a relative way. I think it does not work when sulphide segregation occurs prior to degassing.

Line 119: Replace "S-isotope" by "Sulfur isotope". Please be consistent in using "sulfur" (line: 13) and "sulphur" (line: 25).

Finally, I would recommend to add a short conclusion to highlight the main results and their relevance for future studies in this area.

Manuel Keith

University of Leicester, 19/09/2017

Reviewer #3 (Remarks to the Author):

Comments on the Nature Comm manuscript titled "Degassing-induced, mass-dependent fractionation of sulphur isotopes in hotspot lava from recycled mantle," authored by Beaudry, Longpré, Economos, Wing, Bui and Stix.

The manuscript reports sulfur isotope data of melt inclusions and glasses from El Hierro volcano, Canary Islands. These melt inclusions were previously described by some of the coauthors here, and their compositional variations were shown to represent various degrees of degassing. The main findings are that (1) $\delta^{34}\text{S}$ and S concentration form a positive correlation indicating the degassing sulfur species is isotopically heavier than that of melt, and that (2) the sulfur isotope fractionation during degassing does not result in mass-independent fractionation. The data presented are relatively difficult to obtain as melt inclusion sample preparation is labor intensive and susceptible to sample loss. While I have some reservations regarding the description about $\delta^{34}\text{S}$ calibration, the data and uncertainty assessments are the quality comparable to other SIMS sulfur isotope studies (e.g., Cabral et al. 2013).

As for the assessment of the scientific impact of this study, I must point out that (1) sulfur isotope fractionation during degassing has been described in nature and in laboratory experiments (e.g. ref. 7, 22, 23), and that (2) it is known theoretically and experimentally that chemical reaction yields mass-dependent isotope fractionation (e.g. a textbook by Hoefs, 2009, Stable isotope geochemistry). Therefore, while the data presented are of the interest of the geochemistry community, and the data quality is certainly of the level required for a first-rate publication, I do not see here a transformative discovery often published in the Nature Communications.

As for a minor note, I would like to mention that there is also yet another OIB showing a MIF sulfur isotope variation (Delavault et al. 2016 PNAS). Perhaps, the authors might consider incorporating into the list of reference mentioned in the introduction.

The method section does not describe the exact calibration procedure for $\delta^{34}\text{S}$, especially with respect to compositional variation of standards and samples. There are reports outlining the significant matrix dependent $\delta^{34}\text{S}$ instrumental fraction (e.g., Thomassot et al. 2009, EPSL), I would like to know how this study has treated such issues.

In conclusion, I would love to see the publication of the data presented here. However, from my perspective of a researcher who has studied volatile geochemistry in magmatic melt inclusions, I just cannot find a ground breaking aspect of the data here.

Reviewer #1 (Remarks to the Author):

I only have a couple of questions.

Lines 109-111: it is suggested that the sulfide minerals track the isotope composition of the melt during degassing, but does your data show this? (e.g., Table S1?)

Now lines 234–236: We mean here that the $\delta^{34}\text{S}$ ranges shown by the silicate and sulphide melts are the same, which is well illustrated in Table 1 and Figs. 2a and 3. The new section “Geochemistry of the magmatic sulphides” also explains this more thoroughly (e.g. lines 240-253, 257-260).

138-139: mantle heterogeneities in S isotopes are attributed to “recycling of subducted sulfur,” but what is the source or reservoir of this sulfur? Altered oceanic lavas have negative $\delta^{34}\text{S}$ values (mean = -9‰ Rouxel et al 2008; Alt and Shanks, 2011), so is this sulfate in limestones or sediment pore waters? What about sedimentary sulfide? It seems that some mention of what this sulfur is recycled from is worthwhile here (or refer to a more detailed evaluation of subducted sulfur).

Now lines 327-329: This is a good question, which was initially hard to address in view of the papers mentioned here which looked at the S isotope composition of altered oceanic lavas (Rouxel et al. 2008, Alt & Shanks 2011). More recent evidence (Alt et al., 2013)

suggests that the importance of precipitated seawater sulphate has been underestimated previously, and in particular, the amount of fluid altering the deep oceanic crust may be much more important than previously believed (e.g. Naif et al. 2015, *GGG*), which could be a source of high $\delta^{34}\text{S}$ sulphate in serpentinized peridotite and lower oceanic crust (lines 339-342). We also mention earlier in the article (lines 54-56) the correlation of S and Sr isotopes found for Samoa (Labidi et al. 2015). These authors suggest the recycling of Proterozoic sediments for their enriched $\delta^{34}\text{S}$ values, based on the average $\delta^{34}\text{S}$ of Proterozoic sediments (Canfield & Farquhar, 2009). We make mention of this in lines 338-339.

Reviewer #2 (Remarks to the Author)

General comments

1. The structure of the manuscript suggests that it has previously been submitted to Nature or Nature Geoscience. However, the structure of the manuscript is also its major weakness and it would highly benefit of a clearer structure with sub-sections. For example, the methods are briefly described in the results (lines: 55-60, 73-75). I would recommend to present the methods in an own chapter. Similarly, the last paragraph of the results is interpretation (lines: 75-85) and should therefore be moved to the discussion.

We have added headings for the Results and Discussion sections, each with their own sub-sections. With regards to the Methods, we would like to point out that there was indeed a Methods section in the original manuscript, after the figures (lines 201-271 in the initial manuscript; now 440-510). We agree that the last paragraph of the results was somewhat out of place; it has been removed, and passages from it re-assigned to the first two subsections of the discussion, “Redox evolution of the El Hierro magma” (line 147) and “Sulphur isotope fractionation processes” (line 180). The discussion is divided in five subsections (see below under *Discussion* comments), the titles of which have been removed to comply with the Nature Communications guidelines.

2. Furthermore, the manuscript needs a more detailed description of the occurrence and chemistry of magmatic sulphides. I wonder (1) when the magamtic sulphides formed during melt evolution also with respect to degassing, (2) whether sulphide segregation in this system is a single, multistage or continuous process and (3) the most important question does this have any effect on their S isotope composition? Question (1) and (2) have been discussed in recent studies from Patten (2012 *CanMineral*; 2013 *ChemGeol*) and in our new study (Keith et al., 2017 *ChemGeol*). This would significantly improve the manuscript and helps to support and highlight the interesting and important results and conclusions of this study. Hence, in my opinion the manuscript is generally suitable for publication in Nature Communications after major revisions.

This was an extremely valuable comment, and the new manuscript discusses the sulphides in a lot more detail. In the sub-section of the results “Sulphur isotopes, textures and geochemistry of the sulphides” (line 112), we distinguish various groups of magmatic sulphides based on their geochemical and textural characteristics, and identify their different isotopic trends. The sub-section “Geochemistry of the magmatic sulphides” (line 243) in the discussion goes more in depth regarding their magmatic history. In particular, we identify the different groups as representing different stages in the evolution of the immiscible sulphide melt. To answer the questions: (1) different lines of evidence suggest that the sulphides appear relatively late in the degassing history, e.g. lines 159-162, 249-

250 (also described by Longpré et al. 2017). (2) We envisage a continuous process for sulphide segregation (lines 279–280, 283–285), but sulphides segregated at a late stage should be affected by the decreased S content of the silicate melt (e.g. lines 280–283). (3) All evidence points towards the minimal effect of sulphide segregation on their isotopic composition or on that of the melt. Instead, they acquire their isotopic composition from the degassing and isotopically fractionating melt (e.g. lines 202-210, 233-234, 257-260). The references suggested by the reviewer here were of significant help, as well as a selection of older references from the literature (Czamanske & Moore, 1977, *Bull. Geol. Soc. Am.*, Mungall et al. 2005, *GCA*; Kress 1997, *CMP*; Gaetani & Grove 1999, *EPSL*; Peach et al. 1990, *GCA*; Fleet et al. 1993, *CMP*; Kullerud et al. 1969, *Econ. Geol. Monogr.*). The reviewer’s own paper (Keith et al. 2017, *Chem Geol*), which compares sulphides from arc and back-arc settings to MORB sulphides, was particularly useful: we used their data to submit our own OIB sulphides to a similar analysis and comparison between different tectonic settings. This is shown in Fig. 4 and discussed in lines 288-301.

Specific comments

Lines 18-19: I don’t really like the phrase “magma redox, S speciation and degassing”. I recommend to change it to: “redox conditions, S specification and degassing of hotspot magmas.

We agree with the reviewer and have modified this phrase accordingly (Lines 19–20).

Lines 20-21: This last sentence is not necessary. The previous sentence (lines: 18-19) represents a good conclusion of the study. Based on this last sentence the reader may wonder, why the authors talk about redox, speciation and degassing before but then reduce it to a degassing effect. Does this mean that only degassing is important? I would delete that sentence.

While the initial sentence — “Disentangling the effects of degassing from source heterogeneity is thus critical for understanding the geodynamic implications of S isotopes at volcanic hotspots” — remains one of the main take-aways from the paper, we have moved it to the discussion (lines 318–320) and replaced it in the abstract with one that addresses the nature of the mantle source with respect to the absence of S-MIF (lines 21–23), which was lacking in the initial abstract.

4) Introduction

The introduction lacks any background information about melt inclusions and magmatic sulphides, however, the manuscript focuses on their S isotope composition. A short overview of the recent literature must be added.

We have modified the introduction, in particular by better outlining the issues pertaining to mantle heterogeneities and its various end-member components. We briefly describe melt inclusions and their utility in lines 63–65. Our last introductory paragraph (lines 61–76) does outline the potential importance of magmatic processes in affecting the S-isotope composition of magmas. Our results and discussion sections go more in depth about the nature of silicate melt inclusions and magmatic sulphides.

Line 23: Are these really all geochemical processes and no physical processes? I would rather write: “Chemical and physical processes within the mantle...”.

The fractionation of sulphur isotopes occurs during chemical reactions, but these are indeed affected by physical conditions, so we have incorporated this suggestion in line 25. We also added “biological”, since sulphur is crucial for the metabolism of several microbial organisms, and they produce important S isotope fractionations.

Line 27: Change “mantle circulation” to “mantle convection”.

We prefer using mantle circulation here, since “mantle convection” implies convection cells for the general reader, which may not be an accurate description of the circulation involved in the development and rising of mantle plumes.

Line 28: Based on the title the reader thinks that this paper is about mass-dependent fractionation (MDF) but here you talk about MIF. This has to be clarified. I would change the title as mentioned above and in addition in this part of the introduction you should tell the reader briefly what the difference between MDF and MIF is and what it can or cannot tell you based on previous studies and with respect to the aim of this study

MDF and MIF are both relevant to this paper. The title reflects our observation that the isotopic fractionation in El Hierro sulphides is mass-dependent, which can only be constrained if 3 isotopes are measured. This in itself is an important result, since it contrasts with other studies having measured triple S isotopes in sulphides from OIB and found MIF signatures (Cabral et al. 2013, Delavault et al. 2016). We have improved the second introductory paragraph that now better summarizes the difference between the two than in the initial version (e.g. lines 37–42, 47–60).

Line 28-31: I would not present such an equation in the introduction this should be part of the methods. Just write here that MIF is defined by the $\Delta^{33}\text{S}$ values. If necessary the authors can refer to the methods section for further details

Now lines 37–38. We believe it is necessary to introduce the equations that describe S isotope compositions, since these are relevant for understanding what we refer to in the results and discussion sections. We note that other recent studies published in Nature Communications do have such equations for S isotopes as part of their introduction (e.g. Sansjofre et al. 2016, Li et al. 2016).

Line 32: Be consistent by using “S” or “sulphur”. I would use “S” throughout the manuscript. I know what you mean but how do you get the Archean surface S into the mantle? I assume this is due to plate subduction.

Now line 45. This is indeed due to plate subduction; we have added a comment to that effect for the mantle S cycle (lines 44–46).

Line 45-53: This paragraph should not be part of the introduction. Here you present the geological overview, as well as details which are commonly part of the petrographic and chemical results. Move this into different chapters/sections. Moreover, figures should not be presented in the introduction, these are your results. The introduction has to provide an overview of previous research with respect to the aim of the study

We have added the section “Geological context and sample description” (line 77) after the the introduction, where we give a little bit more detail about the Canary Island mantle. Figure 1 is introduced here for images of the melt inclusions and sulphides.

5) Results

Add a brief geological overview to the manuscript, which sets your samples into a geological contexts. If there are significant differences between different samples it would worth to produce a table to show this. This would help you to keep this chapter short.

See previous comment regarding the geological overview, which we prefer not putting in the results section, since it is not technically a result.
We added Table 1 to summarize the differences between samples.

The manuscript really lacks a petrographic section to present the results based on Fig. 1.

Here the authors should describe the host phases of the melt inclusions and magmatic sulphides. I wonder whether the magmatic sulphides are hosted only in one or in different phases. This is important because in the latter case the data would represent different stages of magma evolution. Furthermore, the authors should estimate the size of the melt and sulphide inclusions. This is important with respect to the beam diameter of the SIMS, please see also my comment below.

The new sections describing the petrography of the sulphides and discussing their geochemistry address this comment. See lines 94 and 98 for description of host phases, and their relationship to isotopic composition on lines 107-109, 130 and Table 1, as well as Supplementary Tables 1 and 2. See lines 110-111 and 143-145 for a brief description of the size of melt and sulphide inclusions, respectively.

Lines 55-60: This should be part of the methods chapter, where the authors then can also define $\Delta^{33}\text{S}$ based on the equation presented in the introduction (line: 30). It makes sense to have some of the methods in the electronic supplement, however, the reader must understand and have the chance to evaluate the data without the supplement. Hence, I highly recommend to add a methods chapter. In addition, in this chapter the authors should tell the reader the general set up of the SIMS, i.e. what beam diameter has been used. This is important with respect to the size of the melt and magmatic sulphide inclusions. I wonder whether the signal was homogeneous or whether it sometimes represented a mixture of the inclusions and the host phase. This should briefly be described in the methods more details can then be shown in the supplement.

Now lines 93-103. We prefer keeping this short overview of the work done in the main text to introduce the results, with a more detailed description of SIMS set-up in the Methods chapter at the end. The mention of the number of inclusions analysed also serves to put emphasis on this unprecedentedly large dataset for S isotope analyses, in particular for the sulphides which all exhibit $\Delta^{33}\text{S}$ of 0 ‰. We did however slightly expand the Methods, in relation to comments by reviewer #3 and provide additional details as Supplementary Information.

Lines 66-68: The authors refer here to Fig. 3 but I can't see the terrestrial mass dependent fractionation line in this diagram. It would be good to show it in the diagram or at least note it in the figure caption.

This was implicit in the last sentence of the caption; however, we have added a parenthesis in line 398 to clarify that the “negative x-direction at $\Delta^{33}\text{S} = 0$ ” is by definition the terrestrial mass-dependent fractionation line.

Line 68-69: The authors should refer to Fig. 2b again, where a homogenous inclusions is shown. Again, this is something which should be described in a petrographic results. According to Fig. 3 (supplement) I wonder why Ni hasn't been mapped because in Fig. 3a (supplement) the authors point out that Ni-rich sulphides occur, what I would expect in a basaltic melt. For the classification of the sulphides I would recommend to produce a triangular plot Fe-Ni-Cu including the stoichiometric compositions of pyrrhotite, pentlandite, cubanite and chalcopyrite. As for example, shown in Patten et al. (2012 CanMineral) and Keith et al. (2017, ChemGeol). This diagram may be part of the supplement.

We carefully addressed this useful comment by drafting new figures (Figs. 1c-f, 4, 5) and more systematically refer to them in the text. We expanded Figure 1 from 2 to 6 panels, and also have expanded Supplementary Figure 3 to 30 panels showing elemental maps for Fe, S, Cu, Ni and Ti of 5 sulphide globules spanning the 3 different groups. A new Supplementary Figure 4 shows an additional sulphide from group 2 with elemental maps for S, Fe, O, Si and Ti, to highlight the presence of oxides in this group. We have also produced the suggested Fe-Ni-Cu ternary diagram (Fig. 4a) showing the compositions of our sulphides in relation to those of other tectonic settings as shown by Keith et al. 2017. This new diagram emphasizes the differences between group 1 and group 2 sulphides, which helps in describing their evolution (also see new Figure 5, showing the relationship between the composition of the sulphides and their S isotopic composition).

Line 70-72: This is interesting so $\delta^{34}\text{S}$ varies between different types of sulphide inclusions. Hence, it is important to figure out when they formed during silicate melt evolution and whether they represent the same evolutionary stage.

We agree. Lines 183-190 now describe this in more detail, as well as the new discussion sub-section on the “Geochemistry of the magmatic sulphides”, in particular lines 252–253, 257-260 and 276-287.

Line 72: I disagree, Cu is in the same liquid as Fe, Cu and S when the silicate melt reaches the S saturation limit and an immiscible sulphide liquid forms. From this immiscible sulphide liquid then first mss (Fe-Ni-rich) exsolve followed by iss (Cu-Fe-rich). Please see Barnes et al. (2006, ContribMineralPetrol) and Patten et al. (2012, CanMineral). This is basically what I mean with presenting some recent results of this area of research in the introduction.

We agree that the initial sentence (“The lowest $\delta^{34}\text{S}$ values are observed for these Cu-bearing and heterogeneous sulphides, consistent with a Cu-sulphide phase exsolving from the magma at a late stage in sulphide precipitation”) was misleading, although the main take-away point here, which was that the Cu-rich sulphides represent a later stage in the evolution of the sulphide melt, still holds true. We have described this order of sulphide crystallization in more detail (lines 260-265), with reference to the experiments of Fleet et al. (1993; *CMP*) on the fractional crystallization of sulphide melt.

Lines 75-85: This is interpretation and has to be moved to the discussion. This part could be used as some introductory section for the discussion.

As mentioned in our reply to general comment #1 above, this paragraph has been removed and reassigned to parts of the Discussion.

Line 77-78: This is exactly what I mean, so timing of sulphide segregation from the silicate melt affects $\delta^{34}\text{S}$. Hence, it is important to distinguish between earlier and later sulphides. That would be a really important result.

Lines 78-80: That's not clear to me, please clarify. Why "however" don't you write above that this is case? Maybe that's me but I struggle here.

The initial sentences on lines 77-80 have been removed, as they were indeed misleading, since we meant that "plausibly" the segregation of sulphide melt could have an isotopic effect on the isotopic composition of the magma, but that evidence suggested otherwise. A new paragraph on lines 198-211 provides a clearer explanation ruling out the isotopic influence of sulphide segregation.

8) Discussion

I recommend to add a few sub-chapters/sections to the discussion. One of them could be introductory based on the last paragraph of the results, which I recommended to move to the discussion. I would also add a section discussing the effect of sulphide segregation at different stages during silicate melt evolution and its effect on the S isotope composition of the sulphides, if this is something that can be shown and discussed

We have substantially expanded the discussion, which we have divided in five subsections, outlined below. As mentioned above, we removed the sub-heading titles to comply with journal guidelines.

- 1) *Redox evolution of the El Hierro magma* (line 147): this first sub-section lays out the background on how the conditions were changing during magma ascent which are the basis for understanding the temporal relationships between degassing and sulphide melt precipitation. This background is necessary to discuss and explain the S isotope data.
- 2) *Sulphur isotope fractionation processes* (line 180): here we first summarize again the main observations, and give theoretical background on the mechanisms leading to S isotope fractionation.
- 3) *Degassing-induced S isotope fractionation modelling* (line 212): this section describes how we quantitatively modelled S isotope fractionation of the El Hierro magma based on melt inclusion data of Longpré et al. (2017), which identifies degassing as the only process able to reproduce our S isotope data. Figure 2a shows the results of this modelling, which nicely encompasses our entire dataset. We also show here that sulphide segregation would not have an observable effect on isotopic composition of the melt, hence the observed range in the sulphides reflects segregation from a melt with changing $\delta^{34}\text{S}$.
- 4) *Geochemistry of the magmatic sulphides* (line 243): Here we explain the evolution of the sulphide melt, supported by new Figures 4-5. It is particularly important to have a consistent model for the co-evolution of sulphide geochemistry and isotopic composition to explain the lower $\delta^{34}\text{S}$ of group 2 and 3 sulphides and assess potential matrix effects during the SIMS analyses (see reply to last comment of R#3 below). A

brief discussion of the sulphide compositions for El Hierro (OIB) compared to other tectonic settings also sets the stage for the last subsection about the Canary Island mantle source.

- 5) *Mantle source and the deep S cycle* (line 302): the last two paragraphs close the loop by going back to the question of the origin of OIB and recycling of subducted S, as discussed in the introduction. They remain essentially unchanged from the initial manuscript, apart from the incorporation of the new results on S-MIF at Pitcairn from Delavault et al. (2016; see 2nd comment of R#3 below), as well as a brief mention of the potential source of positive $\delta^{34}\text{S}$ for recycled S (see reply to 2nd comment of R#1 above).

Line 90: I would replace "...vary in parallel with the oxidation state (as monitored by oxygen fugacity or $f\text{O}_2$)" by "... are representative for the oxidation state of the magma, as monitored by oxygen fugacity ($f\text{O}_2$).

Now line 194: We agree that the word "parallel" is misleading, and have replaced "... vary in parallel" with "... are linked to the oxidation state of the magma, as monitored by $f\text{O}_2$ ", a concept which is also now introduced earlier in the text (lines 152-156).

Line 95: Please be consistent in using supplement or extended data. In Fig. 4 (supplement) please point out that the total pressure equals the $f\text{H}_2\text{O}$. I'm not sure whether this correct (please check) but this is what the authors write above (lines 91-92).

Line 98: Please be consistent in using $f\text{H}_2\text{O}$ or total pressure, or is it not the same?

We have added Supplementary Figure 6 to that effect showing the relationship between P and $f\text{H}_2\text{O}$, as well as a brief comment in the main text relating the two (lines 214-215) and a short discussion in the supplementary material (lines 142-147).

Line 99: What do you mean with "parameterization"? Conditions or set-up?

Now line 222. We have slightly rephrased this sentence. Parameterization refers to the use of parameters to describe a function; the parameterization of Jugo et al. (2010) relates $f\text{O}_2$ to $\text{S}^{6+}/\Sigma\text{S}$ ratios, and that of Iacono-Marziano et al. (2012) relates pressure of melt inclusion entrapment to $\text{CO}_2\text{-H}_2\text{O}$ contents.

Lines 109-111: This could work but you have to know when degassing and sulphide segregation occur during melt evolution at least in a relative way. I think it does not work when sulphide segregation occurs prior to degassing.

Now lines 234-236. Our textural evidence and isotopic data suggests that sulphide segregation did not occur prior to significant degassing (e.g. see lines 156-159, 249-253).

Finally, I would recommend to add a short conclusion to highlight the main results and their relevance for future studies in this area.

We have preferred to keep our ending the way it was, as it touches the geological implications of our results and extends them to a broader scale. Several recent studies published in Nature Communications also do not have a separate conclusion highlighting results, but rather discuss the implications of these results (e.g. Sansjofre et al. 2016, Li et al. 2016, Le Voyer et al. 2017).

Reviewer #3 (Remarks to the Author):

The manuscript reports sulfur isotope data of melt inclusions and glasses from El Hierro volcano, Canary Islands. These melt inclusions were previously described by some of the coauthors here, and their compositional variations were shown to represent various degrees of degassing. The main findings are that (1) $\delta^{34}\text{S}$ and S concentration form a positive correlation indicating the degassing sulfur species is isotopically heavier than that of melt, and that (2) *the sulfur isotope fractionation during degassing does not result in mass-independent fractionation*. The data presented are relatively difficult to obtain as melt inclusion sample preparation is labor intensive and susceptible to sample loss. While I have some reservations regarding the description about $\delta^{34}\text{S}$ calibration, the data and uncertainty assessments are the quality comparable to other SIMS sulfur isotope studies (e.g., Cabral et al. 2013).

As for the assessment of the scientific impact of this study, I must point out that (1) sulfur isotope fractionation during degassing has been described in nature and in laboratory experiments (e.g. ref. 7, 22, 23), and that (2) it is known theoretically and experimentally that chemical reaction yields mass-dependent isotope fractionation (e.g. a textbook by Hoefs, 2009, Stable isotope geochemistry). Therefore, while the data presented are of the interest of the geochemistry community, and the data quality is certainly of the level required for a first-rate publication, I do not see here a transformative discovery often published in the Nature Communications.

Isotope fractionation during degassing has indeed been described previously in nature and experiments, and is not known to produce any mass-independent fractionation. However, no other study before has measured S isotopes simultaneously in melt inclusions, matrix glasses and sulphide inclusions, which allows to track both the S isotope signature of the source and the effect of degassing on the S isotope composition of the magma (lines 69–72). Therefore, our study is novel in that it (1) captures large $\delta^{34}\text{S}$ fractionation caused by degassing in melt inclusions in situ, (2) reconstructs the positive $\delta^{34}\text{S}$ signature of the Canary Island mantle source, and (3) identifies mass-dependent $\Delta^{33}\text{S}$ in an unprecedentedly large number of magmatic sulphides in OIB. The magnitude of isotope fractionation observed and the match between our modelling and data is also unprecedented. Additionally, our samples are extremely S-rich for silicate melts, which raises interesting and important questions with regards to the origin of ocean island basalts, their volatile budget and redox properties.

While ref. 7 mentioned here (Mandeville et al. 2009, now ref. 8) describes large S isotopic fractionation during open-system degassing at Mount Mazama, that magma fractionates towards positive $\delta^{34}\text{S}$, i.e. in the opposite direction as that observed for El Hierro. Furthermore, those measurements were made on bulk rock samples, after chemical extraction of S. Our in-situ measurements by SIMS in melt inclusions provide a much clearer evidence for isotopic fractionation during degassing. We are only aware of three other studies having measured $\delta^{34}\text{S}$ in melt inclusions: Gurenko & Schmincke, 2001 (ref. 6), Black et al., 2014 (ref. 7) and Bouvier et al., 2008, *Journal of Petrology* (not cited). We stress that the trend exhibited by our samples is a much better fit to the expected isotopic fractionation (e.g. Marini et al., 1998; ref. 14) than that observed in those studies.

As for point #2, the majority of S isotope fractionations caused by chemical reactions are of course mass dependent, which we now acknowledge more explicitly in the text (lines 41–42). We incorporated this qualification (“mass-dependent”) to our title because it is of

interest to the mantle geochemistry community, with respect to the recent findings of S-MIF in OIB of the HIMU (Cabral et al. 2013, ref. 4) and EM (Delavault et al. 2016, ref. 5) types, since the Canary Islands are often characterized as having affinities with the HIMU mantle end-member. Our study is only the third one to use SIMS analyses in magmatic sulphides to address this problem, and contrasts with refs. 4–5 by the absence of a MIF signal in the Canary Islands. Yet, the range of $\delta^{34}\text{S}$ values observed in our sulphides is very similar to those of refs. 4–5, hence the possibility that mass-dependent fractionation due to degassing may be superimposed on their mass-independent signatures must be considered, as an alternative to the presumed negative $\delta^{34}\text{S}$ signature for the Archaean protolith suggested by these authors (lines 21–23, 47–49, 313–318). The number of individual sulphides subjected to this cutting edge in situ $\Delta^{33}\text{S}$ analysis in our work (n=49) is also far greater than previously done (n=4 for ref. 4, n=8 for ref. 5).

Therefore, we believe that our combined measurements of S isotopes in melt inclusions, matrix glass and magmatic sulphides are transformative since they allow to (1) track in situ the isotopic evolution of a natural magmatic system to an unprecedented level of detail, and (2) distinguish between the effects of degassing and those of source heterogeneity, which is critical for understanding the geodynamic implications of S isotopes at volcanic hotspots. Our results also open the discussion on the “intermediate” redox character of OIB relative to MORB and subduction zone magmas, and the potential geodynamic links between the three. The new section on sulphide geochemistry also supports this (e.g. Fig. 4b).

As for a minor note, I would like to mention that there is also yet another OIB showing a MIF sulfur isotope variation (Delavault et al. 2016 PNAS). Perhaps, the authors might consider incorporating into the list of reference mentioned in the introduction.

This was a very helpful comment, as we were unaware of these results since that publication came out at the time of writing. We therefore incorporated this reference to the introduction (lines 44, 49) and discussion (line 328), which is interesting as it may also point to older plume sources for Pacific vs. Atlantic OIB (lines 307-313).

The method section does not describe the exact calibration procedure for $\delta^{34}\text{S}$, especially with respect to compositional variation of standards and samples. There are reports outlining the significant matrix dependent $\delta^{34}\text{S}$ instrumental fraction (e.g., Thomassot et al. 2009, EPSL), I would like to know how this study has treated such issues.

We now address this comment in more detail in lines 58-76 of the Supplementary Information. We have also added a table summarizing the instrumental mass fractionation of our sulphide standards.

Thomassot et al. 2009, *EPSL* estimated the proportions of the different sulphide phases present under each spot created by their SIMS analyses to reconstruct the “true” $\delta^{34}\text{S}$ of each analysis using the variable instrumental fractionation bias respective to each phase. However, we note that the instrumental fractionation for pyrrhotite and chalcopyrite was similar, and higher in magnitude than observed in our calibration: our internal calibration standard (Anderson pyrrhotite; $\delta^{34}\text{S} = +1.4 \pm 0.3 \text{ ‰}$, line 10 of Supplementary Information) had measured $\delta^{34}\text{S}$ values on average 2.4‰ lower than its true $\delta^{34}\text{S}$. The standards used by Thomassot et al. 2009 included OPM (chalcopyrite; $\delta^{34}\text{S} = 2.29 \text{ ‰}$) and Enon (pyrrhotite;

$\delta^{34}\text{S} = 0.90 \text{ ‰}$), which both had depletions of 9-12 ‰ between measured and true $\delta^{34}\text{S}$, and were not significantly different between each other across different analytical sessions.

The range of $\delta^{34}\text{S}$ values observed in our study within the compositionally and texturally homogeneous group 1 sulphides (1.0 to -4.0 ‰) is relatively wide, suggesting a fractionation control other than matrix dependence. The group 2 sulphides, which contain various phases (pyrrhotite, cubanite and oxide), reach more negative values with $\delta^{34}\text{S}$ ranging from -1.8 to -7.1 ‰, but the fact that the quenched group 3 sulphides, which have low EPMA totals but otherwise pyrrhotite-like composition (e.g. overlap with group 1 in ternary diagram, Fig. 4a), also have a wide range from -2.3 to -9.6 ‰ supports our claim that the range reflects natural variability, as matrix effects should be negligible between group 1 and 3.

Reviewer #2 (Remarks to the Author):

Dear Patrick and co-authors,

I investigated the manuscript for a second time. I noticed that it has been revised carefully and all major comments by myself and the other reviewers have been satisfactorily addressed.

The manuscript has a clear structure now, which makes it much more reader friendly. For this purpose a detailed Petrography section has also been added, which was one of my recommendations.

Furthermore, a detailed interpretation about the co-evolution of magmatic sulphide chemistry alongside with its S isotope composition has been added, which I also recommended. Published magmatic sulphide data has been used and compared with the newly produced data providing interesting and important new results of the evolution of sulphide liquids in OIB systems. This also includes Fig. 4, which is new and which significantly improved the manuscript.

In summary the revised manuscript is well written and structured, and nicely combines petrographic, geochemical and S isotope investigations in a consistent model about S degassing and S recycling in OIB systems.

I really appreciated the changes, which significantly improved the manuscript. I have no further comments and I recommend to accept the manuscript for publication in Nature Communications.

Apologies for the delay.

Kind regards,

Manuel Keith (14/06/2018)

Reviewer #3 (Remarks to the Author):

I am commenting on the revised submission of the manuscript reporting sulfur isotope data of melt inclusions and glasses from El Hierro volcano, Canary Islands, titled "Degassing-induced, mass-dependent fractionation of sulphur isotopes in hotspot lava from recycled mantle," authored by Beaudry, Longpré, Economos, Wing, Bui and Stix. The studied melt inclusions were previously described by some of the coauthors here, and their new data corroborating compositional variations which were shown to represent various degrees of degassing.

As I have written in my previous comments, I acknowledge that the manuscript presents a quality data set clearly demonstrating the relationship between sulfur content (i.e. degassing) and isotopic composition. For that regard, I agree with the authors that the data shows the most coherent S systematic to date. However, I am still deeply bothered by the way that the authors choose to present the manuscript. Essentially, the manuscript discuss S isotope fractionation during degassing and evolution of sulfide melt (L147-287), while concluding on the geochemical context of the sample itself, such as OIB, recycling, and HIMU source (L302-347). In the end, the paper reports an observation of S isotope fractionation due to a magmatic process, then discuss about the source related issue. In the end, I am not sure what exactly is the main point of the paper.

Therefore, I request the authors to clarify the point of the paper stressing one of two messages:

(1) If the main message of the paper is the discovery of post-Archean recycled oceanic crust (thus non-MIF recycled sulfur) and its potential lithology, the title and abstract should reflect mainly on that point. In such case, the analysis of degassing fractionation is merely a procedure to deduce the source isotopic composition. The abstract clearly presents this message of the mantle recycling, but the title is not.

If author decides to determine the source composition, there should be some efforts to address issues of crustal interaction, and magma mixing. I agree that S isotope variation is dominated by degassing, then are the authors saying that it is not possible to detect isotopic mixing?

(2) If the observation of non-MIF S isotope systematics during magma evolution is the main message, as the title suggests, the abstract needs to stress more the discovery related to degassing process.

To be fair to the authors, I have no issue with individual conclusions/arguments presented in the paper, except for some which need clarifications (listed later). This is way I am just “requesting” here for the clarification of the message.

Here are other passages which require clarification:

L47 “the proposed positive covariation of . . . “ it would be more appropriate to say “the potential positive covariation” or “the possible presence of positive covariation”.

L77 - 91 “Geological context . . .” Unnecessary assertions should be suppressed from this section, notably, “L81 this context thus offers . . .”, “L84 with potential implications for . . .”.

L150 - 151 Having “However” here, logically these should be one paragraph.

L164 “the absence of sulphide in these crystals may also indicate . . . phenocrysts [ref 28]” This is a misleading sentence with unfounded assertion, because ref 28 did not argue for PREFERENTIAL nucleation on Fe-Ti oxide. The ref 28 simply says Fe-Ti oxide provides nucleation cite. Thus, the authors do not have an explanation as for why olivine phenocryst does not contain sulfide inclusion.

L197-198 I see no reason here there is a paragraph break here.

L205-208 “Furthermore, while sulfide . . . “ Without any explanation why the authors reached this statement, it is a merely an assertion. The authors should provide a quantitative argument on why sulfide droplets only represents small fraction. Otherwise, the passage should be suppressed.

L212 Redundant. It has been said already.

L288-301 The discussion in this paragraph is out of place. The passage explains that sulfide composition of El Hierro reflect a high fO₂ magmatic environment, similar to arc than MORB. I do not know why comparison to arc is important here, or even the potential for ore-forming processes. These are not a part of main discussion of the paper, and distracting to readers.

METHOD

Given that the glass standard P1326-2 appear to be highly heterogenous (0.8 +/- 0.1 permil), because of replicate analyses exceeding analytical precision, the nature of the glass standard should be provided. The size, variability of sulfur content, or other volatile content, presence or absence of micro-crystals.

Appendix

Equation 3 There is something wrong with this equation. Wouldn't you agree?

Reviewer #3 (Remarks to the Author):

I am commenting on the revised submission of the manuscript reporting sulfur isotope data of melt inclusions and glasses from El Hierro volcano, Canary Islands, titled "Degassing-induced, mass-dependent fractionation of sulphur isotopes in hotspot lava from recycled mantle," authored by Beaudry, Longpré, Economos, Wing, Bui and Stix. The studied melt inclusions were previously described by some of the coauthors here, and their new data corroborating compositional variations which were shown to represent various degrees of degassing.

As I have written in my previous comments, I acknowledge that the manuscript presents a quality data set clearly demonstrating the relationship between sulfur content (i.e. degassing) and isotopic composition. For that regard, I agree with the authors that the data shows the most coherent S systematic to date. However, I am still deeply bothered by the way that the authors choose to present the manuscript. Essentially, the manuscript discuss S isotope fractionation during degassing and evolution of sulfide melt (L147-287), while concluding on the geochemical context of the sample itself, such as OIB, recycling, and HIMU source (L302-347). In the end, the paper reports an observation of S isotope fractionation due to a magmatic process, then discuss about the source related issue. In the end, I am not sure what exactly is the main point of the paper.

As indicated by the reviewer, the single most important message of our previous manuscript version was somewhat unclear. This is partly because our paper draws **two** important conclusions. (1) We identify and model the strong effect of degassing on $\delta^{34}\text{S}$ in both the silicate and sulphide melts, and conclude that disentangling this effect is key to draw accurate interpretations of the $\delta^{34}\text{S}$ records of volcanic rocks. This first point is somewhat technical, but it is necessary and it will be of broad interest in the community. (2) Our second, "big picture" conclusion is the non-MIF nature (post-Archaean) of S in the recycled oceanic component in the Canary Island mantle plume, which contrasts with recent findings on other OIB. This comes across clearly in our revised title, the introduction and concluding paragraphs of the manuscript.

As previously suggested by reviewers #2 and #3, we felt the old title was somewhat confusing and did not optimally reflect our conclusions. We believe the new title significantly clarifies the messages.

We have also slightly rearranged parts of the discussion for clarity; in particular, the sequence of arguments related to sulphide separation has been reorganized with the following changes:

- (1) The statement on the small contribution of sulphides to the total S budget has been moved to the second paragraph of the discussion (“It is worth noting...”, line 174), where the cause of sulphide separation is discussed.
- (2) The links between textural, compositional and isotopic characteristics of sulphides were previously discussed in two different paragraphs; those sections have now been combined (line 235: “Compositional and textural...” with lines 236–242: “The smooth, homogeneous sulphides...”).

Therefore, I request the authors to clarify the point of the paper stressing one of two messages:

(1) If the main message of the paper is the discovery of post-Archean recycled oceanic crust (thus non-MIF recycled sulfur) and its potential lithology, the title and abstract should reflect mainly on that point. In such case, the analysis of degassing fractionation is merely a procedure to deduce the source isotopic composition. The abstract clearly presents this message of the mantle recycling, but the title is not.

As stated above, this is the main, big picture message: the absence of MIF signal in all magmatic sulphides (despite a large $\delta^{34}\text{S}$ range) contrasts with previous studies of S isotopes in HIMU-type OIB. We have changed the title to better emphasize this discovery: we have removed the term “mass-dependent” which added confusion surrounding the isotopic fractionation process, and replaced it with “post-Archaean” to highlight the contrast with previous studies.

If author decides to determine the source composition, there should be some efforts to address issues of crustal interaction, and magma mixing. I agree that S isotope variation is dominated by degassing, then are the authors saying that it is not possible to detect isotopic mixing?

We have added statements in lines 183–186 justifying why isotopic mixing and assimilation are unimportant in our study. However, we are not saying that, generally, it will not be possible to detect isotopic mixing. This will depend on the specificity of each system, but our data show that the effect of degassing can be dominant and so it should be carefully taken into account in future studies.

(2) If the observation of non-MIF S isotope systematics during magma evolution is the main message, as the title suggests, the abstract needs to stress more the discovery related to degassing process.

As stated above this is our first, more technical conclusion, which comes across in the first part of the new title, in the middle of the abstract, and requires extensive discussion in the text due to its complexity. Our modelling approach adds constraints to the conditions able to produce the observed trend (e.g. Figs. 2,6). The discussion on redox evolution and geochemistry of the sulphides is also important to verify that the degassing process is also a reasonable explanation for the $\delta^{34}\text{S}$ variability of the sulphides, which is accompanied by geochemical changes (e.g. groups 1, 2, 3 discussed in text).

To be fair to the authors, I have no issue with individual conclusions/arguments presented in the paper, except for some which need clarifications (listed later). This is way I am just “requesting” here for the clarification of the message.

Here are other passages which require clarification:

L47 “the proposed positive covariation of . . . “ it would be more appropriate to say “the potential positive covariation” or “the possible presence of positive covariation”.

We agree with the reviewer, and have applied the first suggestion (“potential positive covariation”).

L77 - 91 “Geological context . . .” Unnecessary assertions should be suppressed from this section, notably, “L81 this context thus offers . . .”, “L84 with potential implications for . . .”.

We have removed the second passage: “ with potential implications for the role of hotspot volcanism in Earth’s volatile cycles”. However, we believe the first assertion is helpful, since it makes the link between the Canary Islands and the mantle components discussed in the introduction, explaining why studying sulphur isotopes in this location might be interesting.

L150 - 151 Having “However” here, logically these should be one paragraph.

Agreed, we removed the paragraph break.

L164 “the absence of sulphide in these crystals may also indicate . . . phenocrysts [ref 28]” This is a misleading sentence with unfounded assertion, because ref 28 did not argue for PREFERENTIAL nucleation on Fe-Ti oxide. The ref 28 simply says Fe-Ti oxide provides nucleation cite. Thus, the authors do not have an explanation as for why olivine phenocryst does not contain sulfide inclusion.

It is true that ref. 28 did not argue for preferential nucleation on Fe-Ti oxide; they mention the occurrence of sulphide with flat surfaces against clinopyroxene and Fe-Ti oxide crystals, suggesting that these may act as nucleation sites. Taking this into account and the observation that sulphides in our samples are abundant in both Fe-Ti oxides and clinopyroxene but not in olivine, our interpretation is that sulphides preferentially nucleate on these crystals. However, we have simplified the sentence and removed the citation of ref. 28 to avoid confusion, which makes it clearer that this is our interpretation and not that of ref. 28.

L197-198 I see no reason here there is a paragraph break here.

Agreed, we have removed the break.

L205-208 “Furthermore, while sulfide . . . “ Without any explanation why the authors reached this statement, it is a merely an assertion. The authors should provide a quantitative argument on why sulfide droplets only represents small fraction. Otherwise, the passage should be suppressed.

We agree that without explanation, this argument is flawed. We have added a small section to the Methods (*Estimation of the S budget of magmatic sulphides*) explaining how we came to this conclusion. We preferred adding it to the Methods, as this quantitative argument could be distracting to readers in the main text.

L212 Redundant. It has been said already.

While we discuss in the previous paragraph the relationship between valence state of S and isotopic affinity, we disagree that this sentence is redundant, since it is the first time in the article where we explicitly discuss the gas phase. We believe it is important to make this emphasis here, since it leads us to write the redox reaction involving H₂S and SO₂, which is directly related to our modelling approach and controls the magnitude and direction of isotopic fractionation (Figs. 2,6).

L288-301 The discussion in this paragraph is out of place. The passage explains that sulfide composition of El Hierro reflect a high fO₂ magmatic environment, similar to arc than MORB. I do not know why comparison to arc is important here, or even the potential for ore-forming processes. These are not a part of main discussion of the paper, and distracting to readers.

This paragraph contains final comments on the geochemistry of the magmatic sulphides, by putting them in a larger context, with the comparison to MORB and arc-related sulphides. We believe it is important since it makes the transition between the sulphide geochemistry and the last part of the discussion which addresses the mantle source. In particular, our last paragraph makes the link between arcs and OIB's, such as their higher oxidation state relative to MORB. Therefore, the observation that sulphide geochemistry also reflect the intermediate character of OIB is interesting, and placing their geochemistry in a global context should be valuable to readers interested in sulphide formation. However, we agree that the last sentence (i.e. potential for ore-forming processes) is a bit out of place, and have removed it.

METHOD

Given that the glass standard P1326-2 appear to be highly heterogenous (0.8 +/- 0.1 permil), because of replicate analyses exceeding analytical precision, the nature of the glass standard should be provided. The size, variability of sulfur content, or other volatile content, presence or absence of micro-crystals.

We have added more information on P1326-2 in the Isotope Ratio Mass Spectrometry (IRMS) section.

Appendix

Equation 3 There is something wrong with this equation. Wouldn't you agree?

We believe the equation ($\delta^x S_{V-CDT} = \frac{R_i}{R_{V-CDT}} - 1$) is correct (e.g. Ono 2017, *Annual Reviews of Earth & Planetary Sciences*). Perhaps this comment refers to the absence of a 1000 multiplier; we prefer to exclude it since this multiplier is implicit in the ‰ delta notation.

End of reviews